# Remotely sensed reservoir water storage dynamics (1984-2015) and the influence of climate variability and management at global scale

Jiawei Hou[1], Albert I.J.M. van Dijk[1], Hylke E. Beck[2], Luigi J. Renzullo[1], Yoshihide Wada[3]

[1] Fenner School of Environment and Society, Australian National University, Australia
[2] Department of Civil and Environmental Engineering, Princeton University, United States of America
[3] International Institute for Applied Systems Analysis, Laxenburg, Austria

*Correspondence to*: Jiawei Hou (jiawei.hou@anu.edu.au)

**Abstract.** Many thousands of large dam reservoirs have been constructed worldwide during the last seventy years to increase reliable water supplies and support economic growth. Because reservoir storage measurements are generally not publicly available, so far there has been no global assessment of long-term dynamic changes in reservoir water volume. We overcame this by using optical (Landsat) and altimetry remote sensing to reconstruct monthly water storage for 6,695 reservoirs worldwide between 1984 and 2015. We relate reservoir storage to resilience and vulnerability and investigate interactions between precipitation, streamflow, evaporation, and reservoir water storage based on comprehensive analysis of streamflow from a multi-model ensemble and as observed at ca. 8,000 gauging stations, precipitation from a combination of station, satellite and forecast data, and open water evaporation estimates. We find reservoir storage has diminished substantially for 23% of reservoirs over the three decades but increased for 21%. The greatest declines were for dry basins in southeastern Australia (-29%), southwestern USA (-10%), and eastern Brazil (-9%). The greatest gains occurred in the Nile Basin (+67%), Mediterranean basins (+31%) and southern Africa (+22%). Many of the observed reservoir changes could be explained by changes in precipitation and river inflows, emphasising the importance of multi-decadal precipitation changes for reservoir water storage. Uncertainty in the analysis can come from, among others, the relatively low Landsat imaging frequency for parts of the Earth and the simple geo-statistical bathymetry model used. Our results also show that there is generally little impact from changes in net evaporation on storage trends. Based on reservoir water balance, we deduce it is unlikely that water release trends dominate global trends in reservoir storage dynamics. This inference is further supported by different spatial patterns in water withdrawal and storage trends globally. A more definitive conclusion about the impact of changes in water releases at global or local scale would require data that unfortunately are not publicly available for the vast majority of reservoirs globally.

## 1. Introduction

Globally the number of large reservoirs - dams impounding more than 3 million m$^3$ (ICOLD 2020) - reached 58,713 in 2020 with a combined capacity of more than 10,000 km$^3$ (Chao et al. 2008). By 2015, reservoirs provide 30–40% of global irrigation water requirements, 17% of electricity generated, and various other services, including domestic and industrial

water supply, recreation, fisheries, and flood and pollution control (Maavara et al. 2020; REN21 2016; Yoshikawa et al. 2014). With projected population increase, demand for water and electricity are also expected to increase substantially (Crist et al. 2017; Zarfl et al. 2015). More dams will likely be built to support increased irrigation for food production and to meet energy demand. For example, by 2014, there were 3,700 hydropower dams either under construction or planned worldwide.

The majority of these are in developing countries, particularly in South America, Southeast Asia and Africa (Bonnema et al. 2016; Mulligan et al. 2020; Wang et al. 2021; Zarfl et al. 2015). However, constructing new reservoirs has become challenging due to a shortage of suitable construction sites and remaining 'underdeveloped' water resources, as well as increased recognition of the profound impacts that impoundments have on the local population and riverine ecosystem (Grill et al. 2015; Grill et al. 2019; Lehner et al. 2011; Nilsson et al. 2005).


Adding to the challenge, evidence is emerging that existing reservoirs in some regions have experienced diminished water storage. Recent water supply failures or near-failures have occurred in the US Colorado River Basin since 2000 (Udall and Overpeck 2017), southeast Australia between 2002–2009 (Van Dijk et al. 2013), Barcelona, Spain, in 2007–2008 (March et al. 2013), Sao Paolo, Brazil, in 2014–2015 (Escobar 2015) and Cape Town, South Africa, in 2015–2017 (Sousa et al. 2018).

However, it is unclear if these events are part of a global climate trend or due to local supply or demand changes. The underlying causes are also not necessarily the same in each case: reservoir storage dynamics are the net result of river inflows, net evaporation (i.e., evaporation minus direct precipitation onto the reservoir) and dam water releases to water bodies and users downstream. A change in the balance between these three terms leads to a change in the storage level. There are also interactions. The physical connection between precipitation, streamflow generation and atmospheric moisture

demand creates positive feedbacks in storage volume changes: e.g., assuming the entire water supply system experiences comparable dry conditions, inflows will decrease while net evaporation and downstream demand for water releases for consumptive use will increase. To mitigate this feedback, reservoir operation rules will typically aim to reduce dam releases in response to lowering storage levels. Only a detailed analysis of the water balance of an individual reservoir can conclusively separate the contributions of these three processes to a change in water storage. However, in practice, a loss of

reservoir water storage in the presence of a decrease in upstream or downstream river flows within the river system indicates that reduced precipitation conditions are the most likely cause, whereas the absence of such a precipitation and streamflow decrease, or even an increase, points towards less prudent reservoir operation, possibly in response to increased demand. Therefore, knowledge of temporal trends in reservoir storage and river flow can be combined to interpret whether trends in reservoir water storage are widespread globally, and if so, whether they are likely to be due to changing climate conditions or

due to other factors. For the majority of large reservoirs, operators keep records of releases and estimated storage volume, inflows and net evaporation. Unfortunately, these data are typically not publicly available, for a variety of commercial, logistical, political and security reasons. Probably mainly because of this, so far there has been no attempt at a global assessment of long-term dynamic changes and attribution of trends in water reservoir storage.

Satellite remote sensing has been widely used to measure reservoir water height, extent and storage. Mulligan et al. (2020) developed a global geo-referenced database containing more than 38,000 georeferenced dams and their associated catchments, but without any descriptive features and measurement information. Database for Hydrological Time Series over Inland Waters (DAHITI) (Schwatke et al. 2015) and the U.S. Department of Agriculture's Foreign Agricultural Service (USDA-FAS) Global Reservoirs and Lakes Monitor (G-REALM) (Birkett et al. 2010) are the two most comprehensive

dataset offering global surface water body height variations derived from satellite altimetry, such as Jason-1, Jason-2, Jason-3, TOPEX/Poseidon, and ENVISAT. Several regional and global time series of reservoir water extent have been produced based on MODIS, Landsat or Sentinel-2 imagery (Khandelwal et al. 2017; Ogilvie et al. 2018; Schwatke et al. 2019; Yao et al. 2019; Zhao and Gao 2018). Reservoir volume dynamics can be estimated at either regional or global scale using existing datasets and approaches to derive both height and extent from remote sensing, but this approach is only suitable for a limited

subset number of reservoirs worldwide due to wide spacing of the satellite altimetry tracks (Busker et al. 2019; Crétaux et al. 2011; Duan and Bastiaanssen 2013; Gao et al. 2012; Medina et al. 2010; Tong et al. 2016; Zhang et al. 2014). Messager et al. (2016) estimated the volume of lakes and reservoirs with a surface area greater than 0.10 km$^2$ at global scale using a geo-statistical model based on surrounding topography information. However, these estimates were not dynamic time series, and so do not enhance our understanding of the influence of climate change and human activity on global reservoir storage.


In this study, we combined Landsat-derived surface water extents, satellite altimetry, and geo-statistical models to reconstruct monthly reservoir storage globally for 1984-2015, and examined long-term trends of global reservoir water storage and changes in reservoir resilience and vulnerability over the past three decades. Part of our objective was to determine the extent to which climate variability and human activity each affected global reservoir dynamics over the past

three decades. It is currently impossible to analyse the influence of human activity at global scale directly: there are very few in situ reservoir water release records available publicly, and no hydrological models that can provide reliable estimates. Instead, we consider all climate terms in the reservoir water balance and infer the influence of the remaining unknown term, water releases. First, we investigated trends in precipitation, streamflow and storage at both the reservoir and basin level. If the trends between these variables show similar spatial patterns globally, then this increases the likelihood that climate

variability commonly explains storage changes. Second, we examined the temporal correlation between precipitation, reservoir inflow and storage change to further understand potential causative relationships. Third, beyond reservoir releases, net evaporation is the only other potential loss term, and we examined what fraction of observed trends in storage was attributable to net evaporation. Using the combined insights, we deduced the role of human activity on reservoir storage change, noting that a direct attribution would require in situ records of reservoir water releases. To support our inference, we

analysed the trends of global water withdrawal to discuss whether it could be a significant factor to lead reservoir storage change.

## 2. Data and methods

### 2.1. Data

### 2.1.1 Surface water extent

The Landsat-derived Global Surface Water Dataset (GSWD) (Pekel et al. 2016) provides statistics on the extent and change of surface water at the global scale over the past three decades at a spatial resolution of 30 m. Clouds, cloud shadows and terrain shadows cause errors or missing data for individual months, but Zhao and Gao (2018) developed an automated method to fill gaps in contaminated image classifications and enhance the accuracy and consistency of reservoir surface water extent estimates. They applied this method to produce a monthly time series of surface water extent dataset for 6,817

reservoirs worldwide, based on mapping of the location and high-water mark as contained in the Global Reservoir and Dam database (GRanD) (Lehner et al. 2011). The average Pearson correlation ($R$) between satellite-derived extent and observed elevation or volumes was improved from 0.66 to 0.92 using the algorithm developed by Zhao and Gao (2018). The resulting data are available from 1984 to 2015 and there are 5,917 reservoirs have continuous observations every month over the 32 years. We used this data here as its temporal consistency fits the purpose of this study for long-term trend analysis.

**Table 1** List of the spatial data used in the analyses with source, resolution and temporal coverage of data

| Name and Abbreviation | Temporal Range | Spatial Resolution | Temporal Resolution | Data Source | Notes |
|---|---|---|---|---|---|
| Global Reservoir Surface Area Dataset (GRSAD) | 1984-2015 | 30 m | Monthly | Zhao and Gao (2018) | Surface water extent for 6,817 reservoirs worldwide |
| Global Reservoirs and Lakes Monitor (G-REALM) | 1992-present | N/A | 10-Day | US Department of Agriculture's Foreign Agricultural Service (USDA-FAS) (Birkett et al. 2010) | Near-real-time surface water height anomaly for 301 lakes and reservoirs worldwide |
| eartH2Observe water resources reanalysis | 1980-2014 | 0.25° | Daily/Monthly | Schellekens et al. (2017) | Global surface runoff ensemble mean of eight state-of-the-art global models |
| Multi-Source Weighted-Ensemble Precipitation (MSWEP) | 1979-2015 | 0.25° | 3-Hour/Monthly | Beck et al. (2017) | Global precipitation by merging gauge, satellite, and reanalysis data |
| The Worldwide Water (W3) model | 1980-2014 | 0.25° | Daily/Monthly | Van Dijk et al. (2018) | Global open water evaporation (Priestley-Taylor potential evaporation) |
| Global Reservoir and Dam Database (GRanD) | N/A | N/A | N/A | Lehner et al. (2011) | Global 6,862 reservoir attributes |
| HydroBASINS | N/A | N/A | N/A | Lehner and Grill (2013) | Global watershed boundaries and sub-basin delineations |
| Global sectoral water withdrawal dataset | 1971-2010 | 0.5° | Monthly | Huang et al. (2018) | Global water withdrawal estimates for irrigation, hydroelectricity, domestic, livestock, manufacturing and mining |

### 2.1.2 Surface water height

The US Department of Agriculture's Foreign Agricultural Service (USDA-FAS) provides near-real-time surface water height anomaly estimates every ten days for 301 lakes and reservoirs worldwide. The water surface height product (G-REALM) was produced by a semi-automated process using data from a series of altimetry missions including Topex/Poseidon (1992-2002), Jason-1 (2002-2008), Jason-2 (2008-2016) and Jason-3 (2016-present) (Birkett et al. 2010). The root-mean-square error (RMSE) of G-REALM altimetry data is expected better than 10 cm for the largest water bodies (e.g., Lake Victoria; 67,166 km$^2$) and better than 20 cm for smaller ones (e.g., Lake Chad; 18,751 km$^2$) (Birkett et al. 2010). The advantage of using satellite radar altimeter to measure surface water height is that it is not affected by weather, time of day, and vegetation or canopy cover. The G-REALM data is currently only available for lakes and reservoirs with an extent greater than 100 km$^2$ although observations for water bodies between 50–100 km$^2$ are expected in future.

### 2.1.3 Auxiliary Data

Daily and monthly *in situ* river discharge observations were collated as part of previous research (Beck et al. 2020) from different national and international sources (Table S1). In total, we archived 22,710 river gauging records. Global monthly surface runoff estimates for 1984–2014 were derived from the eartH$_2$Observe water resources reanalysis version 2 (Schellekens et al. 2017), calculated as the mean of an ensemble of eight state-of-the-art global models, including HTESSEL, SURFEX-TRIP, ORCHIDEE, WaterGAP3, JULES, W3RA, and LISFLOOD (for model details refer to Schellekens et al. (2017)). Precipitation estimates were derived from a combination of station, satellite, and reanalysis data (MSWEP v1.1) (Beck et al. 2017). The representative maximum storage capacity reported in the GRanD v1.1 database (Lehner et al. 2011) was used as a reference value to calculate absolute storage changes. The HydroBASINS (Lehner and Grill 2013) dataset was used to define basin boundaries.

### 2.2 Global reservoir storage estimation

In total, 132 large reservoirs had records of both surface water extent and height for the overlapping period 1993–2015. The interpretation of Pearson correlation depends on $p$ value and the number ($N$) of samples. Among these 132 reservoirs, the average number ($N$) of sample (i.e., the monthly pairs of extent and height) is around 166. We used a significance level of $p<0.01$ to determine the corresponding Pearson correlation threshold with the t-test (Eq. S1). The result suggested that the linear relationship is significant when $R$ is above 0.19. In this context, we conservatively considered $R \geq 0.7$ as evidence of strong correlation. Such a strong correlation between extent and height was found for 58 reservoirs (Group A; Fig. 1).. For these, we estimated the height and area at capacity as the maximum observed surface water height and extent, respectively, and calculated reservoir storage volume ($V_o$ in GL or $10^6$ m$^3$) as:

$$V_o = V_c - (h_{max} - h_o)(A_{max} + A_o)/2 \quad (1)$$

where $A_o$ (km$^2$) is the satellite-observed water extent, $A_{max}$ the maximum value of $A_o$, $h_o$ (m) the satellite-observed water height, $h_{max}$ the maximum value of $h_o$, and $V_c$ (GL) the storage volume at capacity. There were 53 reservoirs with a relationship between $A_o$ and $V_o$ for this overlapping period with a Pearson's $R \geq 0.7$. For these reservoirs, $V_0$ was estimated

from 1984 onwards using a cumulative distribution function (CDF) matching method based on $A_0$.

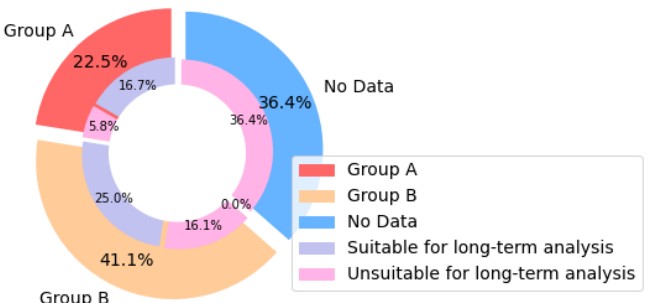

**Figure 1** The total storage capacity in Group A (red) and B (brown) and left unaccounted (blue) and the combined capacity of reservoirs for which the data were suitable (teal) or unsuitable (pink) for long-term analysis.

For 6,611 reservoirs with water extent observations only (Group B; Fig. 1), we used the HydroLAKES method (Messager et

al. 2016) to estimate storage. There are typically two ways to estimate bathymetry based on digital elevation model (DEM) for reservoirs which have no satellite altimetry measurements from space. The first approach is to develop area-elevation curve based on a DEM (Avisse et al. 2017; Bonnema and Hossain 2017). The second method is to extrapolate surrounding topography from the DEM into the reservoir to estimate mean depth (Messager et al. 2016). Although the accuracy of these methods depends on errors inherent in DEM data, the latter one has been proven to a reliable and effective way to estimate

bathymetry of global lakes and reservoirs. A Pearson correlation between predicted and reference depths of $R=0.71$ ($N=7049$) has been reported for global lakes and reservoirs (Messager et al. 2016). Therefore, this geostatistical approach was considered appropriate to estimate reservoir volumes for reservoirs that had only satellite-derived water extent observations.

Messager et al. (2016) proposed a geo-statistical model that provides the empirical relationship of the mean lake or reservoir depth with water surface area and the average slope within a 100 m buffer around the water body. The main assumption of this model is that lake bathymetry can be extrapolated from surrounding topography using slopes. Four empirical equations to predict depth from area and slope were developed by Messager et al. (2016) for different lake size classes (i.e., 0.1–1, 1– 10, 10–100 and 100–500 km$^2$) (Table S2). For each reservoir, water depth dynamics ($D$ in m) from 1984-2015 were

calculated using the surrounding average slope from HydroLAKES and surface water extents (Zhao and Gao 2018) based on the empirical equation appropriate for the reservoir size. In line with Eq. (1), we assumed maximum observed surface water extent ($A_{max}$) as the area at capacity. Water depth ($D_c$ in m) at capacity was calculated as the ratio of volume ($V_c$) and area ($A_c=A_{max}$) at capacity:

$$D_c = \frac{V_c}{A_c} \quad (2)$$

A bias-corrected water depth ($D^*$ in m) was calculated by solving $D$ based on the ratio of water depth ($D_c$ in m) at capacity and maximum observed depth ($D_{max}$ in m):

$$D^* = D \times \frac{D_c}{D_{max}} \quad (3)$$

Storage volume ($V_o$ in MCM) for 1984–2015 was subsequently estimated based on surface water extent ($A_0$ in m) and bias-corrected water depth:

$$V_o = D^* A_o \quad (4)$$

Time series of *in situ* reservoir storage volume measurements are publicly available for a small subset of reservoirs. They can be used to evaluate the uncertainty in the satellite-based storage estimates. Furthermore, data records for some storages can be found in the published literature, derived from grey literature or proprietary data sources. Given the emphasis in trend analysis was on relative changes between the pre- and post-2000 periods, the evaluation of satellite-derived reservoir storage focuses on Pearson's correlation (R) values as a measure of correspondence. In this study, we regard $R$ values ranging from 0.4-0.7 as robust, and 0.7-1 as strong.

### 2.3 Trend analysis and attribution

There are 6,862 reservoirs reported in the GRanD database (Lehner et al. 2011), with the total 6,196 km$^3$ reported storage capacity. In this study, we were able to estimate monthly storage dynamics for 6,695 or 97.6% of the total number of reservoirs, with 3,941 km$^3$ or 63.6% of cumulative capacity (Fig. 1). There were only 58 (0.8%) reservoirs for which storage dynamics could be estimated most directly, by a combination of satellite extent and water level observations (Group A), but together they already represent up to 1,394 km$^3$ (22.5%) storage capacity (Fig. 1). The total capacity of the 172 (2.5%) reservoirs not measured constitutes 2,255 km$^3$ (36.4%) of storage capacity. There were 6,637 (96.7%) reservoirs in Group B for which by the geo-statistical approach could be applied, and their total capacity is 2,547 km$^3$ (41.1%). To ensure consistency in the 1984-2015 time series used for long-term trend analysis, we ignored reservoirs with less than 360 months (i.e., 30 years) of Landsat-derived observations or for which more than five years of water extent observations were inter- or extrapolated by Zhao and Gao (2018). Our focus was on interactions between precipitation, streamflow, evaporation and storage in existing reservoirs, rather than the consequences of new impoundments. Therefore, we excluded from consideration all reservoirs that were destroyed, modified, planned, replaced, removed, subsumed or constructed after 1984. This left 4,573 (66.6%) reservoirs available for with combined storage capacity of 2,583 km$^3$ (41.7%) (Fig. 1).

We calculated linear trends between 1984–2015 in annual reservoir storage, observed streamflow, modelled streamflow, and precipitation for each basin (HydroBASINS Level 3). Trend significance was tested using the Mann-Kendall trend test (p<0.05). The linear trends in modelled streamflow were validated by observed data. We also analysed the correlations between precipitation/streamflow and storage in terms of both time series and linear trend. Net evaporation was calculated for each reservoir as follows:

$$E_n = A(E_o - P) \qquad (5)$$

where $E_n$ (mm) is cumulative monthly net evaporation loss (or gain, if negative), $A$ is reservoir surface area (km$^2$) from Zhao and Gao (2018), $E_0$ (mm) is open water evaporation (Priestley-Taylor potential evaporation from the W3 model (Van Dijk et al. 2018)), and $P$ is precipitation (mm) from MSWEP v1.1 (Beck et al. 2017). The reservoir net evaporation summed for each basin and the ratio of the respective trends in net evaporation and storage were calculated to determine whether the former could explain the latter. Trends in storage and observed streamflow for individual reservoir and river were also analysed to provide additional information about spatial distribution of trends. Unlike the analysis at basin scale above, we do not relate the trend of each individual reservoir to a corresponding river gauge. This is because there is typically a limited number of gauging stations upstream a reservoir, and as such these river flow gauging data cannot accurately represent overall reservoir inflows.

Changes in reservoir resilience and vulnerability between 1984–1999 and 2000–2015 were analysed at the scale of river basins. The reliability, resilience and vulnerability (RRV) criteria can be used to evaluate the performance of a water supply reservoir system (Hashimoto et al. 1982; Kjeldsen and Rosbjerg 2004). The calculation requires that an unsatisfactory state can be defined in which the reservoir cannot meet all water demands, leading to a failure event. *Reliability* indicates the probability that the system is in a satisfactory state:

$$Reliability = 1 - \frac{\sum_{j=1}^{M} d(j)}{T} \qquad (6)$$

where $d(j)$ is the time length of the $j^{th}$ failure event, $T$ is the total time length, and $M$ is the number of failure events. Unfortunately, a single threshold for failure events is not readily determined: firstly, because we did not have access to water demand and release data for each reservoir, and, secondly, because reservoirs are typically operated in response to more than a single threshold. Instead, we assumed that the reliability of each reservoir is designed to be 90%, leaving it in an unsatisfactory state for the remaining 10% of the time. This assumption made it possible to calculate resilience and vulnerability for each reservoir for the assumed 90% threshold. *Resilience* (month$^{-1}$) is a measure of how fast a system can return to a satisfactory state after entering a failure state:

$$Resilience = \left\{ \frac{1}{M} \sum_{j=1}^{M} d(j) \right\}^{-1} \qquad (7)$$

*Vulnerability* (GL) describes the likely damage of failure events:

$$Vulnerability = \frac{1}{M} \sum_{j=1}^{M} v(j) \qquad (8)$$

where $v(j)$ is the deficit volume of the $j^{th}$ failure events. The change in vulnerability was expressed relative to the maximum deficit volume observed. A worked example is shown for the Toledo Bend Reservoir (Texas, USA) (Fig. S5 and Table S5). Four failure events occurred during 1984–2000 and three during 2000–2015. Before 2000, it took an average of three months to recover from failure, with an average deficit volume of 357 GL. After 2000, it took an average of 10.5 months with a larger average deficit volume of 498 GL (Fig. S5). It follows that resilience was reduced (resilience index 0.12 vs. 0.33) and

vulnerability increased (deficit volume 498 vs. 357 GL) when compared to the years before 2000 (Table S5).

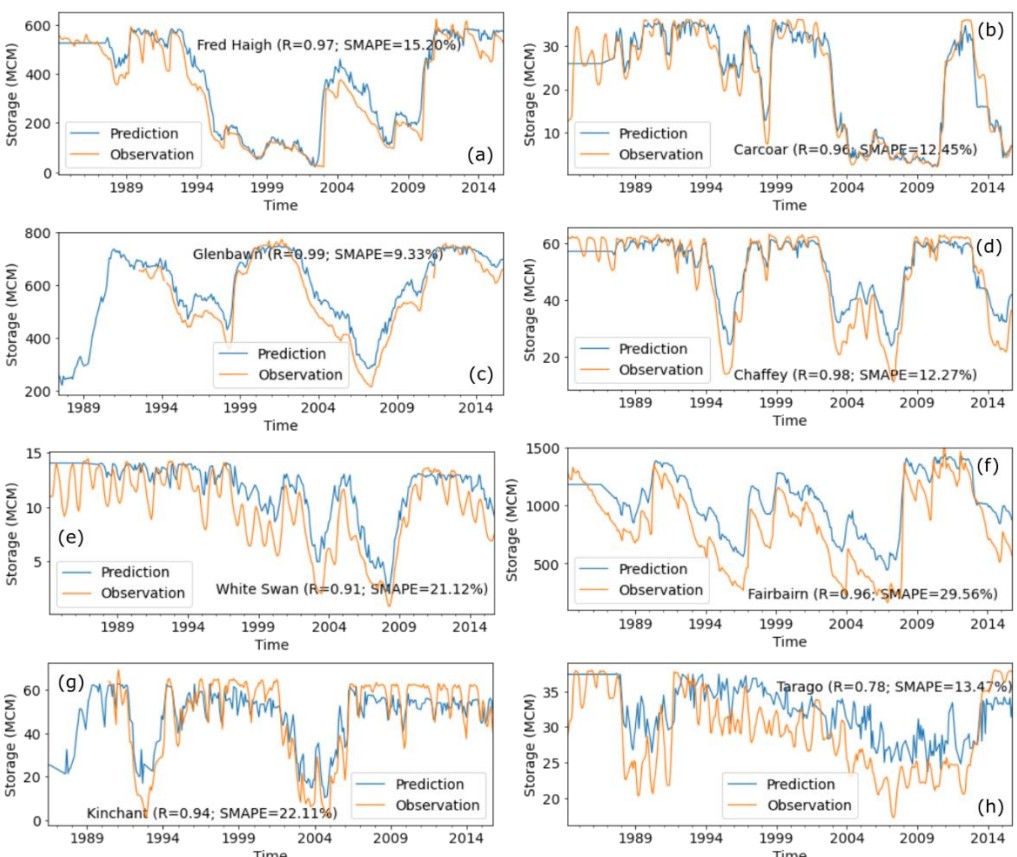

**Figure 2** Validation of monthly reservoir storage time series reconstruction against *in situ* storage data, showing (a, b) robust, (c, d) typical and (e, f) poor results.

## 3. Results

**3.1 Validation of global reservoir storage estimates**

In situ monthly storage records from the US Army Corps of Engineers, US Bureau of Reclamation and Australian Bureau of Meteorology were used for error assessment. There are totally 131 reservoirs with at least 20-year overlapped time series between in situ data and satellite-derived data. We did validation for all these 131 reservoirs (5 for Group A and 126 for Group B). The averaged correlation between observed and estimated volumes is 0.82 ($R≥0.7$ for 82% of the 131 reservoirs).

Messager et al. (2016) reported that the symmetric mean absolute percent error (SMAPE) of the geo-statistical model is 48.8% globally. In our study, the average SMAPE between predicted and reference volumes was 32.13%, lower mainly because we adjusted reservoir storage estimates by reported reservoir capacity. Some cases are shown in Fig. 2. Annual average water levels for Lake Aswan, one of the largest reservoir in the world, were published only as a graph (El Gammal et al. 2010); comparison showed strong agreement between the satellite-derived storage and *in situ* measurements ($R=0.97$,

Fig. S1). In addition, we did cross-validation between Group A and Group B. The results show that 25 of the total 33 overlapping estimated reservoirs show strong agreement ($R≥0.9$) between the two methods, and the average SMAPE between them is 13.1%. This implies good consistency of reservoir storage estimates from Group A and B. Some cross-validation examples are shown in Fig. 3. We investigated the influence of Landsat image quality on the volume time series estimation by comparing time series derived from images with different contamination ratios (0~95%) against the MODIS-

derived lake product (Tortini et al. 2020). The temporal accuracy slightly decreases as the contamination ratio increases (Table S3). However, the overall performance of lake volume estimation using images with contaminated ratio ranging from 5% to 95% is commensurate to using only good-quality images, thanks to the gap-filling method.

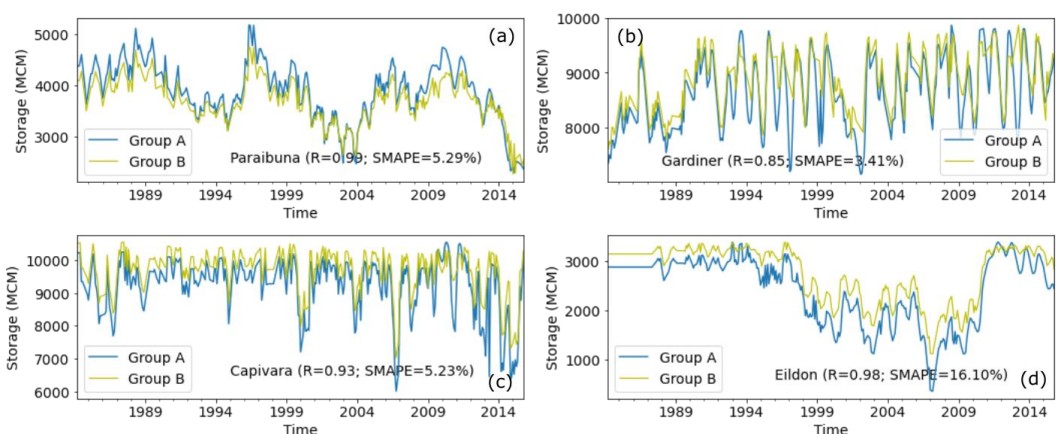

**Figure 3** Validation of monthly reservoir storage time series reconstruction for Group B against results obtained using the method for
Group A, showing (a) robust, (b and c) typical and (d) poor agreement.

### 3.2 Changes in global reservoir storage, resilience and vulnerability

The trends (p<0.05) of water volume dynamics for 4,573 reservoirs and river discharge time series from around 8,000 gauging stations between 1984 and 2015 were analysis here (Fig. 4). We found no systematic global decline in reservoir water availability. Overall, there was a positive trend in combined global reservoir storage of +3.1 km$^3$ yr$^{-1}$, but this was

almost entirely explained by positive trends for the two largest reservoirs constructed before 1984, Lake Kariba (+0.8 km$^3$ yr$^{-1}$) on the Zambezi River and Lake Aswan (+1.9 km$^3$ yr$^{-1}$) on the Nile River (Fig. S2). Reservoir with increasing storage trends are nearly as common as declines. 1,033 reservoirs showed decreasing trends, mainly concentrated in southwest America, eastern South America, southeast Australia and parts of Eurasia, while 944 reservoirs showed increasing trends, distributed in northern North America and southern Africa (Fig. 4a). The global reservoir storage trending pattern is similar

with global river discharge tendency. In particular, a majority of rivers in southwest America, eastern South America, and southeast Australia have reduced river flows (Fig. 4b). There was no apparent relationship between primary reservoir purpose (i.e., irrigation, hydroelectric power generation, domestic water supply) and overall trend, arguably a first tentative indication that climatological influences dominate changes in release management.

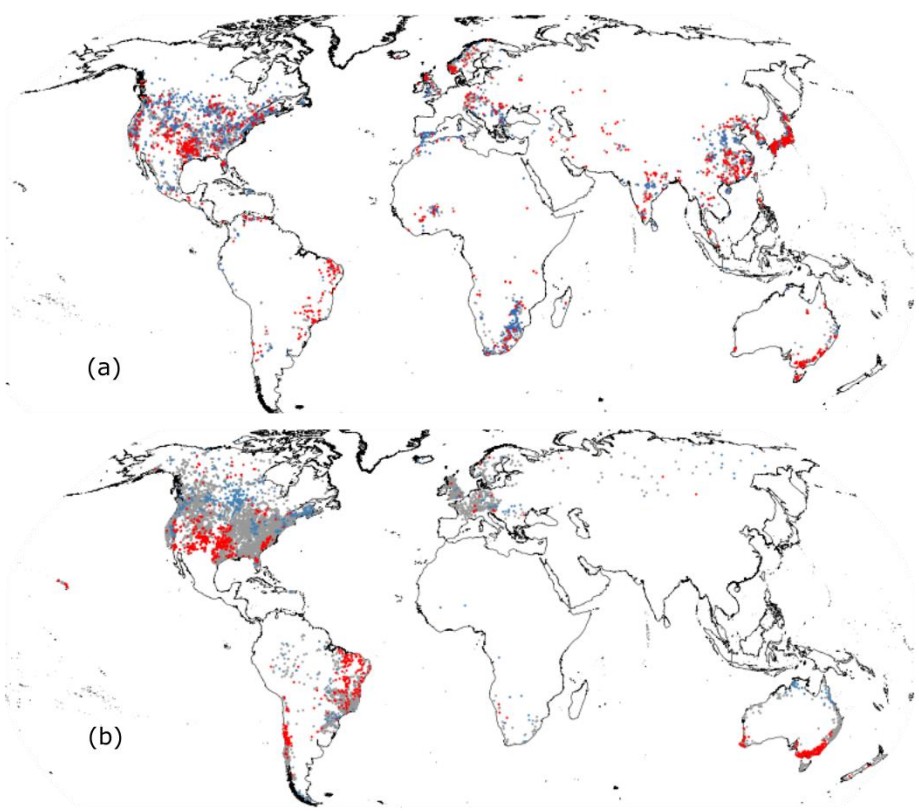

**Figure 4** The trends of storage (a) and observed streamflow (b) for individual reservoir and river globally (p<0.05; increasing: blue; no change: grey; decreasing: red).

The resilience of reservoirs in southwest America (including Mississippi Basin), central Chile, eastern South America, southeastern Australia, the coast of southeastern Africa and central Eurasia have reduced sharply between 1984 and 2015, and the vulnerability of these reservoirs have increased by more than 30% (Fig. 5). In contrast, reservoirs in western

Mediterranean basins, the Nile Basin and southern Africa have stronger resilience and less vulnerability than before (Fig. 5).

All these changes are attributed to changes in reservoir storage, as we found there are a robust positive relationship ($R = 0.64$) between changes from the pre-2000 to the post-2000 period in storage and resilience, and a strong negative relationship ($R = -0.79$) between resilience and vulnerability (Fig. 6). This means that if a reservoir has a decreasing storage, there would be a risk of falling to low capacity more often and enduring larger deficits than before. Increasing storage has the potential to
create other issues, such as overtopping, dam collapse, downstream flooding caused by untimely releases during the wet season, etc. (Simonovic and Arunkumar 2016).

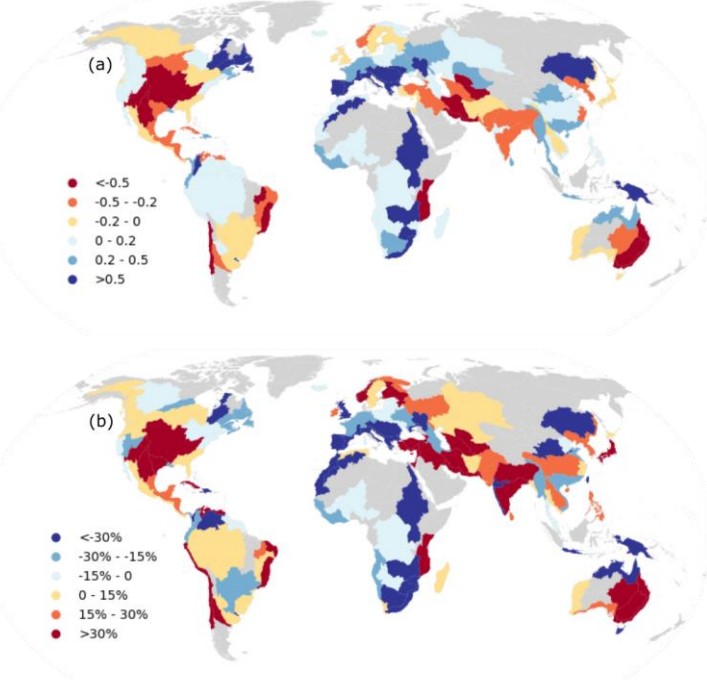

**Figure 5** The change in resilience (a), and vulnerability (b) between pre-2000 and post-2000 (grey shade: no reservoir data).

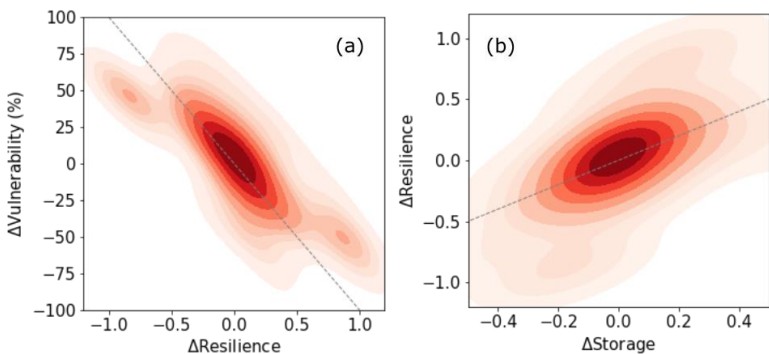


**Figure 6** The relationship (dash grey line: 1: 1 line) between changes from the pre-2000 to the post-2000 period in (a) vulnerability (*ΔVulnerability*) and resilience (*ΔResilience*) and (b) mean storage (*ΔStorage*) and resilience (*ΔResilience*).

### 3.3 Influences of precipitation and river flow on global reservoir storage

We summed storage for individual reservoirs to calculate combined storage in 134 river basins worldwide. Basins losing or gaining more than 5% of their combined storage over the three decades could be found on every continent (Fig. 7c). Among these, 27 (20%) showed a significant decreasing and 37 (28%) a significant increasing trend in reservoir storage (Fig. 7c). If precipitation and runoff trends show the same direction as reservoir storage trends, then it is plausible that climate variations play an important role in reservoir storage trends. On the other hand, if rainfall and runoff show opposite trends to those in reservoir storage, then that could suggest a dominant influence from either net evaporation or water releases. For the majority of the 64 basins, the trends were of the equal sign for storage, runoff and precipitation, suggesting that precipitation changes are commonly the most likely explanation for observed trends (Fig. 7a and b). Opposite trends in precipitation (or runoff) and storage were found for 12 out of 134 basins, with six decreasing and six increasing storage trends. Most of these could be explained by spatial variation within the respective basins (Fig. S3). The linear changes in modelled streamflow were validated against changes in observed streamflow, and the Pearson's correlation between them is 0.77, which indicated modelled streamflow can reliably represent trends in river flow globally (Fig. 8b). There is a robust positive relationship ($R = 0.77$) between linear changes from 1984-2015 in precipitation and streamflow (basin characteristics are assumed largely unchanged in the models) (Fig. 8a). A correlation above 0.6 between them can be found in all these 134 basins except the Niger Basin in Africa and the Parana Basin in South America (Fig. 9b). Linear changes in reservoir storage also have a meaningfully positive relationship ($R = 0.38$, $p < 0.01$, $\rho = 0.51$) with streamflow (Fig. 8c), given the heterogeneous nature of human activities. It means a decreasing trend in streamflow (typically due to precipitation changes) generally leads to a decreasing trend in storage, and vice versa, but not necessarily proportionally. Figure 9a also shows that there are 61 basins that have a robust relationship between annual storage change and inflow with $R$ ranging from 0.4-0.8. They are mainly located in North America, southern South America, Mediterranean, southeastern Australia, and parts of Eurasia. These regions coincide with a large number of measured reservoirs (Fig. 4a) and a large total number of Landsat images over three decades (Pekel et al. 2016; Wulder et al. 2016), and vice versa. The overall relationship between reservoir storage and inflow might therefore be expected to be stronger if more reservoirs were measured and more useable Landsat imagery was available for those basins lacking them in our present analysis. We also found that changes in net evaporation accounted for well below 10% of the overall trends in storage for each of those 64 basins, reflecting that net evaporation rarely explains more than a few per cent in observed storage changes (Fig. 10). In summary, we did not find evidence for widespread reductions in reservoir water storage due to increased releases.

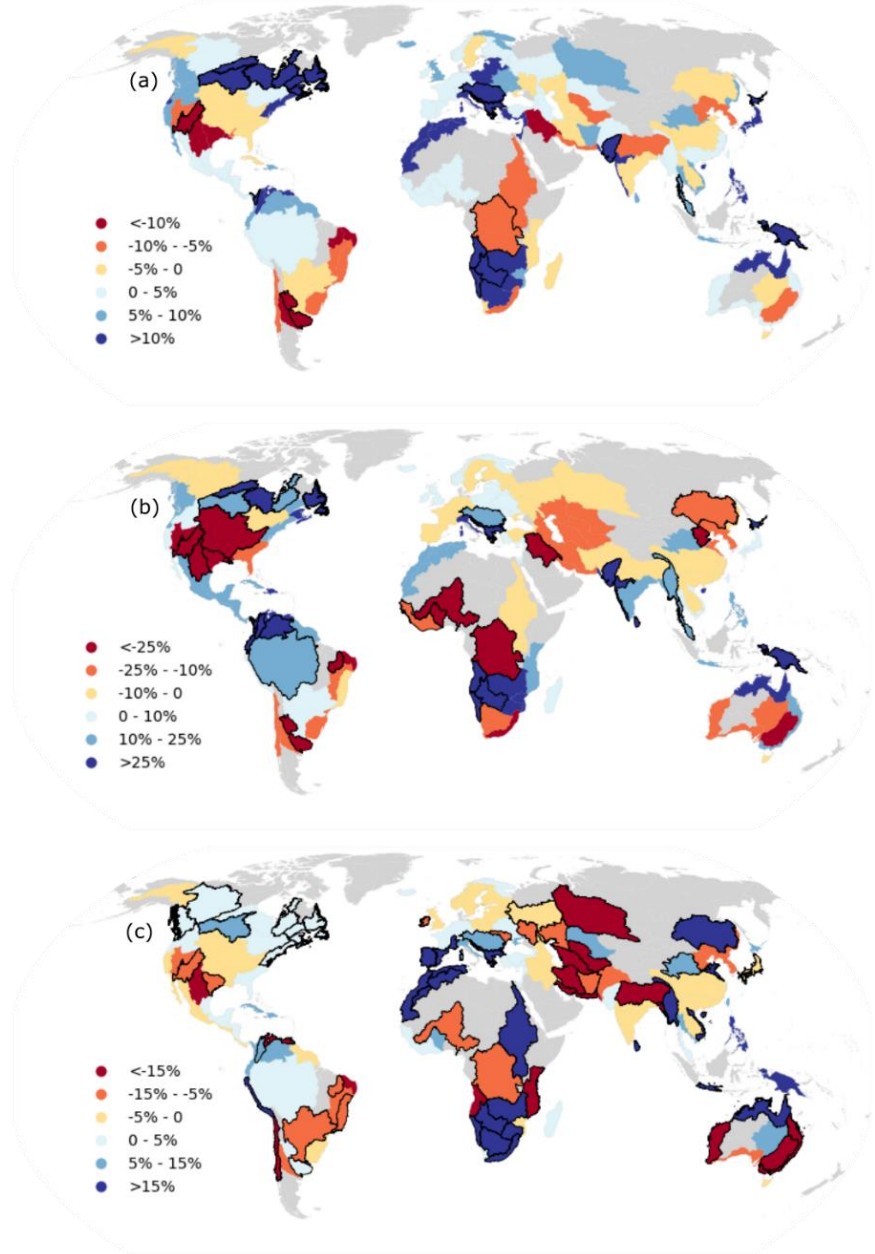

**Figure 7** Linear trends in annual, basin-average (a) precipitation, (b) simulated streamflow and (c) reservoir storage between 1984–2015 (grey shade: no reservoir data; black outlines: trend significant at p<0.05).

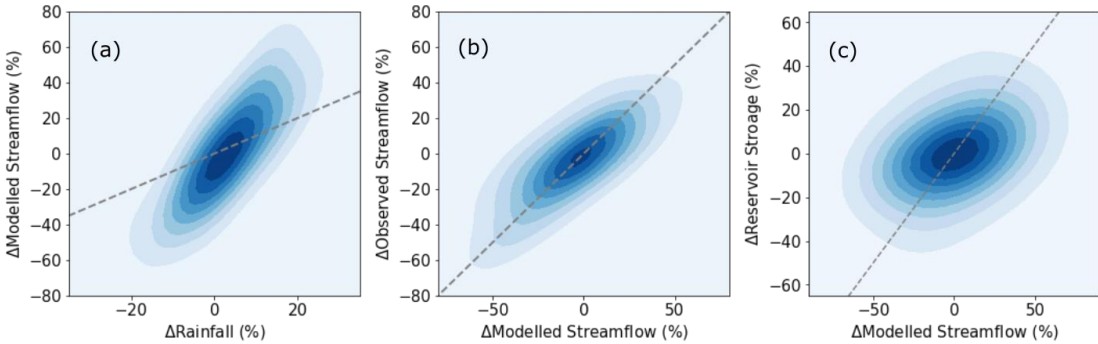

Figure 8 The relationship (dash grey line: 1: 1 line) between linear change from 1984-2015 in (a) annual precipitation (*ΔRainfall*) and modelled streamflow (*ΔModelled Streamflow*), (b) observed streamflow (*ΔObserved Streamflow*) and modelled streamflow (*ΔModelled Streamflow*) and (c) reservoir storage (*ΔReservoir Storage*) and modelled streamflow (*ΔModelled Streamflow*).

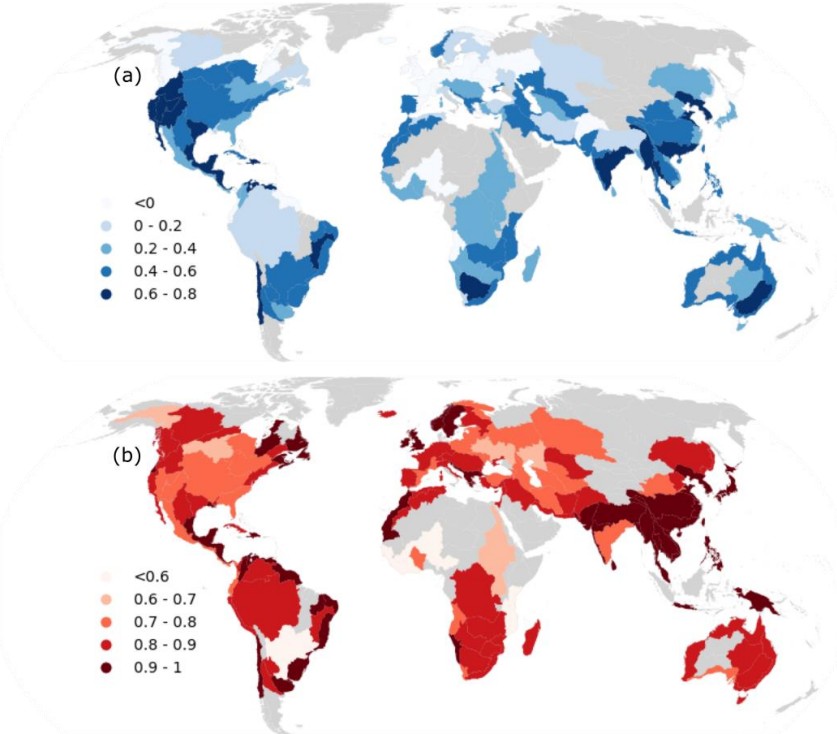

Figure 9 The correlations of annual storage change and reservoir inflow (as approximated by basin modelled streamflow) (a), and reservoir inflow and precipitation (b) in each basin.

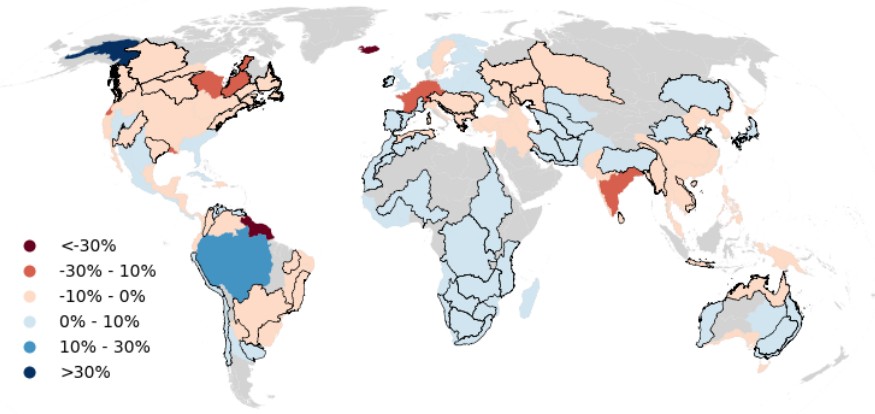

**Figure 10** The ratio of the linear trends in net evaporation and in storage in each basin.

The greatest storage gains occurred in the Nile Basin (+67%), western Mediterranean (+31%) and southern Africa (+22%), and were attributed to very high inflows during 1996-2008, 2008-2010 and 1996-2000, respectively (Fig. S4). Substantial

decreases were found for arid to sub-humid basins in southeastern Australia (-29%), southwestern USA (-10%) and Brazil (-9%) (Fig. 11). Both simulated and observed river discharge data show similar trends and explain the observed storage declines (Fig. 4 and Fig. 7). During Australia's Millennium Drought (2001-2009) (Van Dijk et al. 2013), river flows in the Murray-Darling Basin fell to about half that for 1984–1999 (Fig. 11a), causing a halving of combined storage, before recovering due to high inflows during 2009-2011. In the southwestern USA, three distinct dry periods occurred (Fig. 11b).

Sharp decreases in river flow after 2011 in eastern Brazil led to the lowest reservoir storage levels, with combined losses of almost 18% in 2015 (Fig. 11c). Reservoirs in these basins with reduced storage also predominantly showed reduced resilience and increased vulnerability (Fig. 5).

## 4 Discussion

This study reconstructed monthly reservoir water storage dynamics from 1984-2015 at global scale based on satellite-derived

water extent (Zhao and Gao 2018) and altimetry measurements (Birkett et al. 2010). Where no altimetry data were available, geo-statistical models (Messager et al. 2016) were applied to satellite-derived water extent for reservoir water volume estimation. About a quarter (22.5%, including most large reservoirs) of total reported cumulative reservoir capacity (Lehner et al. 2011) around the world was measured by combining satellite-derived extent and height, while 41.1% was estimated based on geo-statistical models using remotely sensed surface area. There does not appear to be any systematic global

decline in global reservoir water availability, but we found significantly decreasing trends in reservoir water volumes in southeastern Australia, southwestern USA and eastern Brazil, creating the risk that storages fall to low capacity more often (i.e., weakened resilience) and endure larger deficits (i.e., higher vulnerability).

To understand the influence of the reservoir size distribution on the total basin storage trends, we compared the trend directions of total storage in all reservoirs, the top-three largest reservoirs, and the remaining small reservoirs, respectively. We did this for 42 basins with more than 20 reservoirs (4,003 reservoirs in total). Combined storage in these three groups all showed the same trend direction in 27 (62.8%) of basins. The trend in the combined storage for all reservoirs had the same direction as that for the largest few reservoirs for 8 more basins, and the same direction as the combined remaining smaller reservoirs for another 8 basins. This indicates that the largest reservoirs do not always dominate combined total storage dynamics.

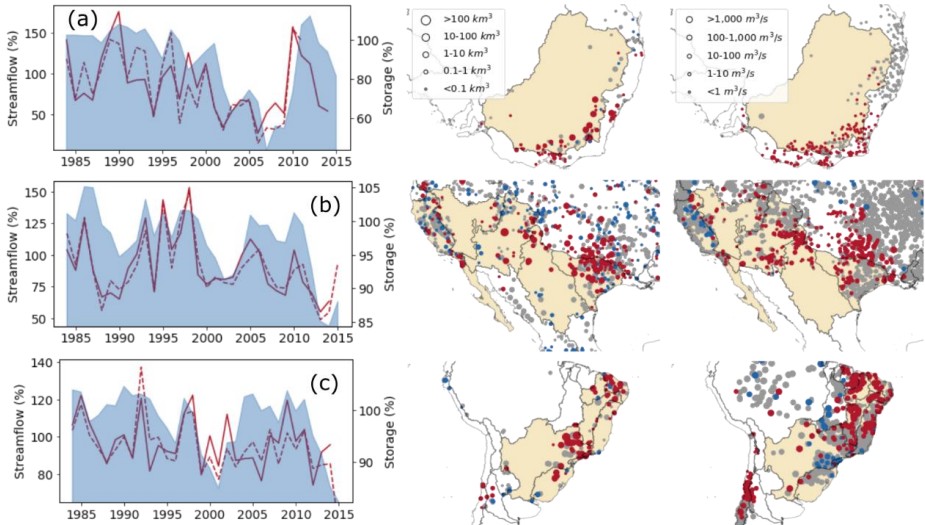

**Figure 11** Time series (left column) of annual combined storage (blue shaded) along with simulated (solid) and observed (dashed line) streamflow, indexed to the reference period 1984–1999, and trends in storage (middle column) and observed streamflow (right column) during 1984–2015 (p<0.05; increasing: blue; no change: grey; declining: red). Shown are (top row) southeastern Australia, (second row) southwestern USA, and (third row) Brazil.

Trends in reservoir storage and river flow showed spatial consistency at both individual and basin scales globally. There was reasonably strong temporal correlation between precipitation, streamflow and storage. Changes in net evaporation only accounted for a small fraction of reservoir volume changes. Mady et al. (2020) and various other authors found that evaporative losses can account for much of the loss of water from small reservoirs (e.g., <0.1 km$^2$) in semi-arid regions. However, this does not necessarily mean that trends in evaporation can explain trends long-term trends in storage, especially for the mostly larger (and deeper) reservoirs considered here. Reservoir storage dynamics ($\Delta V$) are the net result of river inflows ($Q_{in}$), net evaporation ($E_n$) and dam (demand-related) water releases ($Q_{out}$) as:

$$\Delta V = Q_{in} - E_n - Q_{out} \quad (9)$$

We found that $\Delta V$ responds primarily to $Q_{in}$ and that $E_n$ does not seem to have affected $\Delta V$. This indicates dam (demand-related) water releases ($Q_{out}$) are less likely to be the main driver of storage changes ($\Delta V$). Evidence that the impact of human activity is less than that of climate variability is also found in other recent studies. For example, Wang et al. (2017) found that climate variability was the dominant driver of the decreasing trend in lake area across China's Yangtze Plain; human activities only accounted 10-20% of trends despite construction of the Three Gorges Dam upstream. Furthermore,

Gudmundsson et al. (2021) demonstrated that climate change dominates changes in river flow from 1971-2010 worldwide, rather than water and land management.

There is currently no global hydrological model capable of estimating the impact of historical operational water management at the reservoir or basin level with meaningful accuracy. However, to get an indication of the potential impact of human

activity and associated reservoir water releases on reservoir storage changes, we analysed the global water withdrawal estimates produced by Huang et al. (2018). The gridded monthly withdrawal time series for 1971-2010 were spatially and temporal downscaled from 5-year temporal resolution estimates from FAO AQUASTAT and USGS, which were based on national assessments and surveys (Huang et al. 2018). Their estimates provide separate water withdrawal estimates for irrigation, hydroelectricity, domestic, livestock, manufacturing and mining, respectively. The withdrawals are from

reservoirs, rivers and groundwater, and as such cannot be compared directly to reservoir water release, but may provide useful context. We calculated total withdrawals from the six sectors combined and examined trends from 1984-2010 at basin scale. The results show that significant increasing trends in withdrawals in 78 basins, mainly in South America, Africa and Asia, and significant decreasing trends in 29 basins in Europe, Australia, and parts of Northern America, but noting that the magnitude of withdrawals varied widely compared to, for example, total river inflows or reservoir capacity (Fig. 12; Fig.

S6). The global pattern in water withdrawals trends is different from the spatial patterns in precipitation, inflow, and storage (Fig. 7). We calculate that (either significant or non-significant) water withdrawal trends are associated with about equal numbers of increasing and decreasing water storage trends (Table 2). By contrast, rainfall and inflow trends lead to a change in storage in the same direction for around 80% of basins (Table 2). These observations further support the notion that climate trends rather than water withdrawals are primarily responsible for the observed trends in reservoir storage.

Nonetheless, there are basins where storage trends may have been influenced by water withdrawals. For example, inflows increased by 43% in northern Venezuela while total reservoir storage deceased by 15%, conceivably because water withdrawals tripled from 1984-2010 (Fig. 7 and 12). A comparable scenario also occurred in coastal basins in Angola, Mozambique, Tanzania and Kenya. Storages in Iran, Turkmenistan and northern India decreased by an average 33%, which may be attributed to an unknown combination of reduced inflows (-6% - -21%) and increased withdrawals (+42% - +50%),

although it is noted that a large fraction of withdrawals is from groundwater in some of these basins.

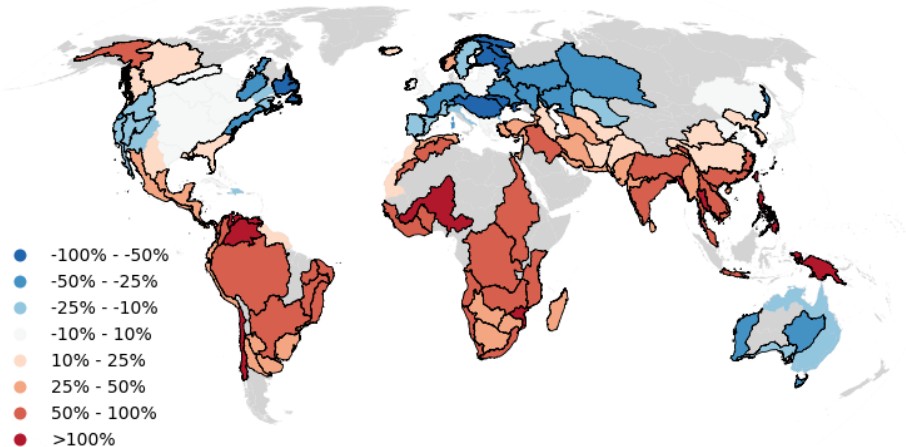

**Figure 12** Linear trends in annual water withdrawal between 1984–2010 (grey shade: no reservoir data; black outlines: trend significant at p<0.05). Note that the magnitude of withdrawals varies strongly between basins.

**Table 2** Comparison of trends in reservoir storage reconstruction against climate variability and human activities.

| Drivers | Reservoir Storage | | | | |
| --- | --- | --- | --- | --- | --- |
| | Trend (number of basins) | Significant increase | Significant decrease | Increase | Decrease |
| Water withdrawal | Significant increase (78) | 18 | 9 | 36 | 42 |
| | Significant decrease (29) | 6 | 6 | 12 | 17 |
| | Increase (93) | 21 | 10 | 43 | 50 |
| | Decrease (41) | 10 | 10 | 19 | 23 |
| Climate (precipitation and inflow) | Significant increase (23) | 11 | 0 | 21 | 2 |
| | Significant decrease (14) | 2 | 4 | 5 | 9 |
| | Increase (70) | 31 | 6 | 54 | 16 |
| | Decrease (64) | 6 | 21 | 11 | 53 |


Accurate temporal pattern estimates were the main purpose in this study because relative water storage and long-term change are more relevant information for water resources management. Our validation results show that 82% of the reservoirs evaluated show strong correlation ($R \geq 0.7$) with water volume measured in situ. In terms of absolute value, water volume estimates were bias-corrected by representative maximum storage capacity from GRanD (Lehner et al. 2011) by assuming that the maximum observed surface water extent coincides with the area at full capacity. Biases remain in some reservoirs due to uncertainties in this maximum storage capacity. Representative maximum storage capacity values reported in GRanD were collected from different sources in the following order of priority: reported maximum or gross capacity, reported normal capacity and reported live or minimum capacity. These uncertainties in reported maximum capacity may have influenced our results for individual reservoirs. This could be solved easily if more accurate reservoir storage or capacity data were available.

The uncertainties and limitations of reservoir storage estimates are mainly from the errors in satellite altimetry data and satellite-derived water extent data. The quality and accuracy of these altimetry measurements depend on the size and shape of water body, surrounding topography, surface waves, major wind events, heavy precipitation, tidal effects, the presence of ice and the position of the altimeter track (Birkett et al. 2010; Busker et al. 2019). The RMSE of water level estimations of a narrow reservoir in steep terrain will be many tens of centimetres (Birkett et al. 2010; Schwatke et al. 2015). DAHITI altimetry data, with RMSE between 4-36 cm for lakes (Schwatke et al. 2015), should have similar accuracy as G-REALM, although its water level observations have so far received less evaluation. The classifier used to produce GSWD surface water data performed quite well, with less than 1% commission error and less than 5% of omission error (Pekel et al. 2016). But no-data classifications in GSWD data caused by cloud, ice, snow, and sensor-related issues could lead to large data-gaps in time series and underestimation of actual reservoir extents (Busker et al. 2019). In general, a no-data threshold is applied to monthly GSWD data for removing imagery with large percentage of contamination before deriving lake and reservoir water extent. It helps reduce the issue to some extent, but contaminated imagery would still remain in the rest of GSWD data. Zhao and Gao (2018) developed an automatic algorithm to repair contaminated Landsat imagery. This has increased the number of effective images by 81% on average and produces continuous reservoir surface area dynamics.

The higher hypsometric correlation we used, the less uncertainties volume estimations would have (Crétaux et al. 2016). We selected correlation threshold of 0.7 in this study, which is lower than Tortini et al. (2020) (R≥0.85) and Busker et al. (2019) (R≥0.9), but higher than Gao et al. (2012) (R≥0.5). The selection of an appropriate correlation threshold can also depend on the purpose of the study. Tortini et al. (2020), Busker et al. (2019) and Gao et al. (2012) aimed to provide accurate measurements for an individual reservoir. Here, our priority is to understand the 32-year volume trend at basin scale. The uncertainties from the individual hypsometry (0.9≥R≥0.7; total 29 reservoirs) therefore average out by temporal (i.e., annual) and spatial (i.e., basin) aggregation.

The total number of Landsat images over North America, southern South America, southern Africa, central Eurasia, and Australia over the past three decades is much larger than in the rest of the world, and particularly in tropical regions (Pekel et al. 2016; Wulder et al. 2016). Regions with sparse Landsat observations can have additional uncertainties in their long-term trend analyses, although this issue has been mitigated to some extent by the approach from Zhao and Gao (2018). In principle, the inflow of sediments into reservoirs could contribute to decreasing storage. However, Wisser et al. (2013) showed that sedimentation caused a total decrease of global reservoir water storage of only 5% over a century (1901 to 2010), and hence we expect the effect of sedimentation on our 32-year analysis to be small. There are studies showing higher sedimentation rates (e.g., Syvitski et al. 2022), so the impact of sedimentation on reservoir trend analysis cannot be discounted entirely. Thus, decreasing storage volume could be exacerbated by sedimentation, while increasing storage volumes could potentially be (partly) explained by it.

Regional storage trends in the dam reservoirs found here are consistent with trends reported in a previous study for 200 lakes (including a few reservoirs) across North America, Europe, Asia and Africa during 1992–2019 (Kraemer et al. 2020). Both lakes and reservoirs are influenced by changing inflow and net evaporation in response to climate variability. Although human regulation has more influence on reservoirs than on natural lakes, our results suggest that for the majority of basins natural influences dominate human impacts, although human impacts on the hydrological regime still exist, of course. For example, Cooley et al. (2021) found that human interventions have resulted in larger seasonal variability in reservoirs than that in lakes globally. In line with the study carried out by Kraemer et al. (2020), we also found that the distribution of global lake and reservoir storage or level long-term trends does not fully reflect the "wet gets wetter and dry gets dryer" paradigm that some have predicted to occur due to anthropogenic climate change (Wang et al. 2012). Reservoirs in dry regions, such as southwest America, southeastern Australia and central Eurasia, have indeed seen deceasing combined storage, while these in wet regions, such northern North America, have increasing storage. However, at the same time we found increasing storage in dry southern Africa and decreasing storages in wet southeastern South America. Additionally, total terrestrial water storage (i.e., the sum of groundwater, soil water and surface water) derived from GRACE satellite gravimetry for the shorter period 2002-2016 showed decreases in endorheic basins in Central Eurasia and the southwestern USA and increases in Southern Africa consistent with our storage changes (Wang et al. 2018).

Reservoir storage dynamics are the net result of river inflows, net evaporation and dam water releases. We found a reasonably strong relationship between changes in river flow and reservoir storage, while changes in net evaporation do not seem to have affected storage trends significantly. We infer that reservoir water releases are unlikely to be the dominant driver of the three-decadal trends in reservoir storage for the majority of basins. However, we acknowledge that this particular conclusion is based on deductive rather than observational evidence, and would benefit from corroboration for any individual reservoir using actual release records, which often exist but not publicly available. Although there are no water demand and supply or dam operation data available globally that could serve as direct evidence, there have been local studies. For example, reservoir operating rules (i.e. reservoir outflow) were inferred from a combination hydrologic modeling and satellite measurements for the Nile Basin, the Mekong Basin, northwest America, and forested region of Bangladesh (Bonnema and Hossain 2017; Bonnema et al. 2016; Eldardiry and Hossain 2019). It was not possible to apply the techniques used in these studies at global scale because of the resulting uncertainties in inferred reservoir inflows. To distinguish the respective influences of human activity and climate variability on reservoir dynamics, greater collaboration and public sharing of *in situ* data on reservoir storage, water release and downstream water use would be required. In some basins, satellite-derived upstream and downstream river discharge dynamics (Hou et al. 2020; Hou et al. 2018) and changes in irrigation area or evaporation (Van Dijk et al. 2018) may be able to provide additional information to better understand the drivers of reservoir water security. The algorithm from Zhao and Gao (2018) could in principle be used to calculate reservoir surface water extent time series beyond 2015, but is reliant on the availability of Landsat-derived GSWD (Pekel et al. 2016).

Such data could also be derived from MODIS or Sentinel 2, and help understand how reservoir water storage change from 2015 onwards. The new NASA Surface Water and Ocean Topography (SWOT) satellite mission should also provide new opportunities to cover a larger number of reservoir ($> 250$ m$^2$) with both surface water extent and height observations for storage estimations (Solander et al. 2016).

## 5. Conclusions

We reconstructed monthly storage dynamics between 1984-2015 for 6,695 reservoirs using satellite-derived water height and extent. For reservoirs with water extent data only, storage was estimated from the surrounding topography using a geo-statistical model. This approach introduces uncertainty but is inevitable as lake bathymetry data based on surveys are typically unavailable, at least in the public domain. The estimated reservoir storages dynamics show strong correlations with averaged $R$=0.82 against publicly available observed storage volume estimates for several reservoirs in the US, Australia and Egypt. Based on the developed global dataset, we found that reservoir storage changed significantly in nearly half of all basins worldwide between 1984–2015, with increases and decreases similarly common and mostly explained by corresponding precipitation and runoff changes. Increases appeared slightly more common in cooler regions and decreases more common in drier regions. With lower-frequency observations, Landsat may not always have fully or accurately captured the storage variability for each reservoir, which can have had an effect on trend analysis. We provided four lines of evidence to explore which factor (precipitation, net evaporation, or dam (demand-related) water releases) drives the global reservoir storage trends. First, we found trend consistency between precipitation, streamflow and reservoir storage. Second, we found robust temporal correlation between precipitation, streamflow and reservoir storage. Third, we inferred the role of human activity based on the reservoir water balance equation: because we found changes in net evaporation only accounted for a small fraction of reservoir volume changes, together with the first two lines of evidence, we can infer that dam (demand-related) water releases are less likely to be the main driver of storage changes. Fourth, we examined water use data and did not find that increasing water use corresponded to deceasing reservoir storage, or vice versa, in the majority of basins. Therefore, we conclude that reservoir volume changes are dominated by (multi-decadal) precipitation changes. Changes in reservoir water storage appear to be predominantly determined by periods of low inflow in response to low precipitation. Future changes in precipitation variability are among the most uncertain predictions by climate models (Trenberth et al. 2014). Therefore, a prudent approach to reservoir water management appears the only available means to avoid water supply failure for individual river systems.

**Data availability:** Global reservoir surface area dataset (GRSAD) is available from the Gao Research Group, Texas A&M University (https://ceprofs.civil.tamu.edu/hgao/pages/models_data.html). Surface water level lake products are courtesy of the NASA/USDA G-REALM program and can be found at https://ipad.fas.usda.gov/cropexplorer/global_reservoir/. GRanD (http://globaldamwatch.org/grand/), HydroBASINS (https://hydrosheds.org/page/hydrobasins) and HydroLAKES

(https://www.hydrosheds.org/page/hydrolakes) products were developed by Global HydroLAB, McGill University. *In situ* reservoir storage data were collected from Australian Bureau of Meteorology (http://www.bom.gov.au/waterdata/), US Bureau of Reclamation (https://www.usbr.gov/uc/water/hydrodata/) and the US Army Corps of Engineers (http://www.nwd-mr.usace.army.mil/rcc/projdata/projdata.html).


**Author contribution:** JH and AIJMVD conceived the idea. AIJMVD, HEB, LJR and YW guided the study. JH carried out the analysis and wrote the manuscript with contributions from all the co-authors.

**Competing interests.** The authors declare that they have no conflict of interest.


**Acknowledgments:** This study was supported by the ANU-CSC (the Australian National University and the China Scholarship Council) Scholarship. Calculations were performed on the high-performance computing system, Gadi, from the National Computational Infrastructure (NCI), which is supported by the Australian Government. We also thank Prof. Bernhard Lehner of McGill University for his feedback on an earlier version of this paper.

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
