# Peer review of "Remotely sensed reservoir water storage dynamics (1984-2015) and the influence of climate variability and management at global scale"

_Hydrology and Earth System Sciences, 2021_

## Author Comment (AC1)

**Response to Reviewer #1 Comments:**

In this manuscript, Hou et al. estimated water storage dynamics for more than 6,000 reservoirs worldwide from 1984 to 2015 using a combination of Landsat imagery, radar satellite altimetry, and geostatistical modeling. They also analyzed the patterns of increasing and decreasing trends globally. Finally, they attributed reservoir storage changes to climate and human variables and found that precipitation and river inflows largely dominated reservoir storage changes.

I feel this is a very interesting study. Previous studies provided long-term storage changes for only dozes of reservoirs. It is really great to see a global dataset of more than 6,000 reservoirs, as compiled in this study. Their attributions on the reservoir storage changes can potentially inform local to regional water resources management. However, I have some major concerns on the quality of the global dataset and the methodology that they applied to attribute the storage changes.

**We thank the reviewer for the detailed and valuable comments and suggestions, which will enable us to greatly improve the quality of our manuscript. Below please find our response to reviewer's comments in detail.**

R1C1) The Landsat satellites does not provide global coverage in the 1980s and maybe in the 1990s as well (Murray et al 2019). The authors did not acknowledge this limitation while stating they quantified reservoir storage from 1984 to 2015 globally. Is the produced storage time series consistent through 1984 to 2015? Could you provide a figure documenting the number of observations in each year in the time series from 1984 to 2015?

"Murray, N. J. et al. The global distribution and trajectory of tidal flats."

**We agree that Landsat-derived products have limitations on providing observations in the 1980s. But this issue predominantly occurs in Oceania, Siberia, Greenland and parts of central and eastern Asia (Pekel et al. (2016); https://www.nature.com/articles/nature20584/figures/5). Landsat-derived water observations are available from 1984 onwards for most parts of Northern America, South America, Africa, Europe, and western and eastern Asia.**

**Furthermore, Zhao and Gao (2018) developed an algorithm to fill gaps in time series when the contamination/occultation in a Landsat image is between 5-95%, and applied interpolation and extrapolation for the missing monthly area estimates (i.e., no images or >95% invalid data). As a result, in their reservoir area product, there are 5,917 reservoirs that have Landsat observations every month from 1984 - 2015 (Fig. 1).**

**Despite that, we were still cautious in using the reservoir area data for long-term storage trend analysis. First, we removed reservoirs for which over five years of data were inter- or extrapolated. Second, we filtered out reservoirs with observations for less than 360 months (30 years), e.g. in New Zealand. After these steps, we found that 4,573 reservoirs constructed before 1984 have consistent Landsat observations and these were used for long-term analysis, compared to the 6,690 reservoirs for which we produced monthly storage dynamics.**

**We will better explain these steps in the revised manuscript.**

R1C2) While this study produces storage changes for a greater amount of reservoirs globally, I do not think the authors fully addressed the limitations that prevent previous studies from documenting reservoir storage dynamics with a better spatial coverage. The authors estimated storage changes for the 132 large reservoirs with both water areas and levels without assessing their consistency. Without a high correlation between water areas and levels, it makes no sense to me to combine these two to deduce storage changes. The authors need to refer to existing studies (e.g., Busker et al.) on quality control before simply combining satellite observations. The authors used a geostatistical method to estimate the storage changes in the vast majority of reservoirs, on which I have an even greater concern. The authors need to be aware that the mean depth, as archived in the HydroLakes dataset, is a ratio of the total volume and maximum lake area. The mean depth does not provide any meaningful information of the actual water depth. Additionally, the geostatistical model adopted by Messager et al. is a spatial model measuring the relationship between the total storages and maximum areas for a large group of water bodies. The authors tried to use the outcome (e.g., mean depth) to estimate storage changes in each individual reservoir, which differs from the purpose of the Messager et al. Unless the authors provide a comprehensive validation, I am not convinced the proposed method is feasible to estimate storage changes for the majority of studied reservoirs here.

"Busker, T. et al. A global lake and reservoir volume analysis using a surface water dataset and satellite altimetry. Hydrol. Earth Syst. Sci. 23, 669–690 (2019)."

"Messager, M. L., Lehner, B., Grill, G., Nedeva, I. & Schmitt, O. Estimating the volume and age of water stored in global lakes using a geo-statistical approach. Nat. Commun. 7, (2016)."

**We thank the review for this suggestion. Following comments R1C2 and R1C7, we will increase correlation (R) thresholds between A-L and between A-V for reservoir storage estimation. We regard R values above 0.7 as indicating strong correlation, and will use this as the correlation threshold. For group A, we will only calculate reservoir storage when the correlation between A-L is above 0.7. Storage dynamics between 1984-1993 (when altimetry data is not available) will be estimated using A if the correlation between A-V is above 0.7 between 1993-2015. We will update the subsequent long-term analysis as well. We can confirm that these stricter measures do not affect the conclusions of our study, however.**

**We would like to clarify that we did not directly use the mean depth archived in the HydroLAKES dataset (Messager et al., 2016). Indeed, this value is not related to the geo-statistical model and is simply the ratio of the reported volume and lake area. Rather, the geo-statistical model provides the empirical relationship of the mean depth with water surface area and the average slope within a 100 m buffer around the water body (Table S2; Supplementary Material). Messager et al. (2016) have validated the predicted lake depth and volume derived from the geo-statistical model against observed data. The symmetric mean absolute percent error (Eq. (1)) and correlation between predicted and observed lake depth are 47.4% and 0.71,**

respectively (Messager et al., 2016). Furthermore, the SMAPE and correlation between predicted and reference volume are 48.8% and 0.95, respectively (Messager et al., 2016).

$$SMAPE = 100 \times \frac{1}{N} \sum \frac{|\text{observed value} - \text{predicted value}|}{(\text{observed value} + \text{predicted value})/2} \quad (1)$$

**We used this statistical model to estimate reservoir depth and volume dynamics from 1984-2015. We will clarify how we used the geo-statistical model for storage estimation in the revised manuscript. In addition, also responding to comments R1C2 and R1C13, we will include the absolute error (SMAPE) in Fig.2 and Fig.3 (L205-210) and list the SMAPE and correlation metrics (Table 1 and 2) for individual reservoirs in supplementary material.**

R1C3) The presented attribution on reservoir storage changes seems to be so simplified that I have many concerns. First, the authors simply compared the directions of the trend in reservoir storage versus that in potential drivers but the analysis only produces coincidence rather than causation. Second, the authors conclude that the evaporation did not significantly impact the reservoir storage but the calculation for the evaporation is too cheap. The authors may need to use more advanced approaches (e.g., Zhao and Gao) in order to draw a confident conclusion. Third, reservoirs, particularly large ones as documented in GranD dataset, are highly regulated by humans. The authors depend on the outputs of global models on estimating human water release from reservoirs. Are the data really reliable for producing trend in human release for each reservoir? In sum, the authors need to pay careful attention to these limitations that potentially affect their conclusions.

"Zhao, G. & Gao, H. Estimating reservoir evaporation losses for the United States: Fusing remote sensing and modeling approaches. Remote Sens. Environ. 226, 109–124 (2019)"

**We thank the reviewer for this comment, but in fact we explicitly considered the difference between coincidence and causation in our study. In a first step, we indeed looked at the coincidence of trends. We identified that the spatial distribution of trends of storage and in situ river flow show very similar global patterns (Fig. 4; L225-227). We could not relate each individual reservoir to a corresponding river gauge because the limited number of gauging stations upstream of reservoirs cannot accurately represent overall reservoir inflows. In a second step, therefore, we performed trend analysis using modeled river flow (validated against in situ river flow in Fig.8b; L272-275) at the basin scale, given total basin water storage can be expected to respond to a change in overall precipitation and streamflow. We confirmed the same directions of trends between precipitation, streamflow and reservoir storage in most basins globally, though not all (Fig. 7; L269-271). Third, we focused more on attribution by calculating Pearson correlations among the different variables, which obviously can provide evidence for, but not proof of, a causative relationship. Thus, we showed that there are reasonably strong correlations among linear trends in precipitation, streamflow and reservoir storage (Fig. 8a and c; L272-275). Furthermore, positive relationships between annual time series of storage change and reservoir inflow and between reservoir inflow and precipitation were found in a majority of**

basins globally (Fig. 9; L276-278). We will attempt to clarify the logic of our analysis in the revised manuscript.

With regards reservoir evaporation, we believe our estimates are robust. Zhao and Gao (2019) estimated evaporation losses for 721 reservoirs in the contiguous United States using three different meteorological datasets, including TerraClimate, North American Land Data Assimilation System phase 2 (NLDAS-2) forcing and Global Land Data Assimilation System Version 2 and Version 2.1 (GLDAS-2 and GLDAS-2.1). We used their monthly reservoir evaporation amount (1000 $m^3$/month) to analyse the trends in net evaporation and compared the trends with the ones derived from the W3 model (Van Dijk et al. 2018) for 721 reservoirs. The results show strong agreement in derived linear trends, especially with regard to the more detailed TerraCliamte dataset (Fig.2). Various hydrological variables estimated by the W3 model have been evaluated in previous studies. Therefore, we argue that the $E_0$ derived from the W3 model are entirely appropriate to analyze linear trends in net evaporation.

The eartH2Observe model estimates do not include the impacts of reservoirs on river flows, and there is currently no global hydrological model that is capable of estimating the impact of historical operational water management at the reservoir or basin level with any acceptable degree of accuracy. Instead, we focused on all relevant variables for the reservoir water balance and tried to infer the influence of the remaining unknown variable (i.e. reservoir releases). Specifically, we did analyse the interaction between precipitation/streamflow/evaporation and reservoir volume and inferred the influence of human activity given that the water volume dynamics in a reservoir is the net balance of inflow (streamflow, affected by precipitation), net evaporation (i.e., evaporation minus direct precipitation) and reservoir releases (L306-312). We will clarify the approach to infer the role of human activity in reservoir storage long-term changes in the revised manuscript.

Specific comments:

R1C4) Line 25: "The majority of …particularly in South America, Southeast Asia and Africa". The authors may consider add more relevant references here.

"Wang, J. et al. GeoDAR: Georeferenced global dam and reservoir database for bridging attributes and geolocations. Earth Syst. Sci. Data 0–52 (2021)"

"Mulligan, M., van Soesbergen, A. & Sáenz, L. GOODD, a global dataset of more than 38,000 georeferenced dams. Sci. Data 7, 1–8 (2020)."

**Thanks. We will include these two references.**

R1C5) Line 65: Schwatke et al. 2019 is another study on estimating long-term lake area changes.

Schwatke, C., Scherer, D. & Dettmering, D. Automated Extraction of Consistent Time-Variable Water Surfaces of Lakes and Reservoirs Based on Landsat and Sentinel-2. Remote Sens. 11, 1010 (2019)

**We will include this paper in the revised manuscript.**

R1C6) Line 89: It is hard to understand "coefficient of determination" here. Could you define or explain it?

**As we used Pearson correlation throughout this paper, we will convert coefficient of determination ($R^2$) to Pearson correlation (R).**

R1C7) Line 120: I do not quite understand what's the purpose showing the correlation between A0 and calculated V0 (based on A0). It makes more sense to me to show the correlation between A0 and h0 as these two are independent estimates. The authors may only need to consider a pearson' R greater than 0.8 (or R2 higher than 0.6) as correlation between A-L or A-V should be pretty high, otherwise it indicates substantial uncertainty in the data sets.

**Thank you for this suggestion. We will increase correlation (R) thresholds between A-L and between A-V for reservoir storage estimation. Please see our response to R1C2 for full details (the first paragraph).**

R1C8) Line 135: the equation does not make sense to me. The authors need to show more details about the rationale.

**We apologize for the confusion. We will clarify the rationale of this equation.**

R1C9) Line 150: "Only 132 reservoirs with both area and level observations….". Do you conclude based on the 132 reservoirs or all reservoirs, majority of which do not have both observations?

**We performed a long-term trend analysis for 4,573 reservoirs, including both Group A (have area and level observations) and Group B (have only area observations). We will modify these sentences to clarify this information.**

R1C10) 164: It seems the MSWEP v1.1 may not be the latest version of the dataset.

**Although there is a latest version of MSWEP now, we carried out our study using MSWEP 1.1 two years ago. We will consider using the latest MSWEP product, but among us are authors of the MSWEP product and with knowledge of the changes between successive versions we do not expect any important impact for the type of long-term analysis done here.**

R1C11) 192: The authors only validated on 1% of the studied reservoirs and the validation samples are located in U.S. only, which could be a concern.

**We provide validation of our calculated storage volumes for 65 reservoirs with publicly available storage data from US and Australia and cross-validation between two volume estimation methods for 33 reservoirs globally. Please see our response to R1C2 for full details (the second paragraph). We note that the availability of such ground data is limited, which was the primary reason for us to develop a remote sensing-based methodology. However, to further demonstrate the validity of our storage estimates, we compared our product with MODIS-derived water**

storage dynamics from 1992 to 2018 for another 100 reservoirs from Tortini et al. (2020) in L202-204.

R1C12) 194: What do you mean by "published"? The authors use pearson's R (correlation) for doing validation, which does not give insights on the accuracy of estimated values.

**We will change "published" to "observed". We think Pearson correlation is the most important validation metric for this study as we focused on trend analysis and this depends on temporal pattern rather than absolute value. However, we will include the absolute error (SMAPE) valuation and compared this validation to Messager et al. (2016). Please see our response to R1C2 for full details (the second paragraph).**

R1C13) Figure 2: it would be more clear to show global-scale evaluation and move the evaluations on individual cases into the supplementary.

**Thank you for this suggestion. We will show the validations for individual reservoirs in supplementary material. Please see our response to R1C2 for full details (the second paragraph).**

R1C14) Line 214: "a positive trend in combined global reservoir storage of 3.1 km3 per yr". This rate seems to less than 10% of earlier estimates on global reservoir storage rates (e.g., Chao et al.). Could you provide an uncertainty estimate for this rate?

"Chao, B. F., Wu, Y. H. & Li, Y. S. Impact of artificial reservoir water impoundment on global sea level. Science (80-. ). 320, 212–214 (2008)."

**We explicitly performed trend analysis for the reservoirs constructed before 1984 only (L150-152), to remove the influence of new reservoir water impoundments from 1984-2015. This was done to provide a clearer understanding on the interaction between precipitation/streamflow/evaporation and reservoir volume. Our study differs from Chao et al. (2008), who focused on cumulative storage by increased water impoundment. We will clarify this in the revised manuscript.**

R1C15) Line 215: "this was almost entirely explained by positive trends for the two largest reservoirs in the world, Lake Kariba (+0.8 km3 yr-1) on the Zambezi River and Lake Aswan". This statement is confusing. I know some completed projections of megadams in China and Brazil, such as the three gorges dams.

**Please refer R1C14. Many mega-dams in China and Brazil were constructed after 1984 and have not been included in our long-term analysis. We will clarify this in the revised manuscript.**

R1C16) Line 219: "while 948 reservoirs showed increasing trends, distributed in northern North America and southern Africa". The reported hotspots of increasing reservoir storage are inconsistent with the patterns of recent dam booms.

**Please refer R1C14. We did not consider dams constructed after 1984 for long-term analysis. We will clarify this in the revised manuscript.**

R1C17) Figure 4: This map is confusing to me. For example, China may be the global lead in dam constructions during the study period. Why its reservoir storage decreased? Is the data correctly shown in this map?

**Please refer R1C14. The map shows trends for reservoirs constructed before 1984. We will clarify this in the caption of Fig.4.**

R1C18) Line 245: "We summed storage for individual reservoirs to calculate combined storage in 134 river basins worldwide". Do reservoirs show a similar pattern of storage change in the same river basin? Is it more meaningful to analyze each of them individually?

**A majority of reservoirs in each basin shows the same trends where there is a significant trend in total storage (Fig. 11; L301-305). We showed the trend analysis at both the individual (Fig.4a; L225-227) and basin scale (Fig.7c; L269-271). Due to the limited abilities of the hydrological model to simulate inflow for individual reservoir, we performed our climate analysis at the basin scale instead, given total basin water storage can be expected to respond to the change in overall precipitation and streamflow.**

R1C19) Line 268: "In summary, we did not find evidence for widespread reductions in reservoir water storage due to increased releases". Reservoir storage increase could be a result of increased impoundments. Did you consider that?

**Please refer R1C14. We have removed the effect of the increased reservoir water impoundments from 1984-2015. Therefore, reservoir storage increase cannot be a result of increased impoundments.**

R1C20) Line 339: As Zhao and Gao used contaminated Landsat imagery to increase the monthly coverage of reservoir areas by 80%, do the estimates from poor-quality images affect your storage analysis? I know some studies (e.g., Busker et al) only adopted good-quality images due to this issue.

"Busker, T. et al. A global lake and reservoir volume analysis using a surface water dataset and satellite altimetry. Hydrol. Earth Syst. Sci. 23, 669–690 (2019)."

**We think the consistency of observations from 1984-2015 is more important for long-term analysis. This is the reason why we used the reservoir area product developed by Zhao and Gao (2018). In addition, Zhao and Gao (2018) demonstrated that the correlations between observed and estimated reservoir areas were improved from 0.66 to 0.92 by 'repairing' contaminated Landsat images.**

**Table 1** The SMAPE and Pearson correlations between predicted and reference volumes for 65 reservoirs.

| Grand ID | Dam Name | Latitude | Longitude | Capacity (MCM) | R | SMAPE (%) |
|---|---|---|---|---|---|---|
| 307 | Fort Peck Dam | 48.00 | -106.41 | 23560 | 0.98 | 28.58 |
| 597 | Glen Canyon | 36.94 | -111.49 | 25070 | 0.99 | 39.12 |
| 753 | Garrison Dam | 47.51 | -101.43 | 30220 | 0.97 | 30.98 |
| 870 | Oahe Dam | 44.46 | -100.40 | 29110 | 0.97 | 30.48 |
| 6199 | Darwin River Dam | -12.83 | 130.97 | 265 | 0.90 | 8.20 |
| 6579 | Tinaroo Falls | -17.16 | 145.55 | 407 | 0.91 | 11.05 |
| 6581 | Paluma | -18.95 | 146.15 | 12.3 | 0.77 | 18.28 |
| 6582 | Copperfield River Gorge | -19.04 | 144.12 | 20.6 | 0.79 | 14.54 |
| 6583 | Ross River | -19.41 | 146.74 | 417 | 0.95 | 59.49 |
| 6586 | Peter Faust | -20.37 | 148.38 | 500 | 0.94 | 28.74 |
| 6588 | Burdekin Falls | -20.65 | 147.14 | 1860 | 0.89 | 14.70 |
| 6592 | Eungella | -21.14 | 148.39 | 131 | 0.94 | 27.14 |
| 6593 | Kinchant | -21.21 | 148.90 | 62.8 | 0.94 | 22.11 |
| 6594 | Fairbairn | -23.65 | 148.07 | 1440 | 0.96 | 29.56 |
| 6595 | E.J. Beardmore | -27.91 | 148.65 | 101 | 0.84 | 30.51 |
| 6600 | Windamere Dam | -32.73 | 149.77 | 368 | 0.96 | 26.89 |
| 6603 | Carcoar Dam | -33.62 | 149.18 | 35.8 | 0.96 | 12.45 |
| 6605 | Wyangala | -33.97 | 148.95 | 1220 | 0.97 | 22.22 |
| 6613 | Burrinjuck | -35.00 | 148.60 | 1026 | 0.92 | 39.02 |
| 6618 | Blowering | -35.40 | 148.24 | 1628 | 0.92 | 42.00 |
| 6619 | Googong | -35.42 | 149.26 | 124.5 | 0.97 | 7.52 |
| 6620 | Bendora | -35.45 | 148.83 | 11.1 | 0.37 | 12.17 |
| 6621 | Corin | -35.54 | 148.84 | 75 | 0.72 | 19.99 |
| 6629 | Eucumbene | -36.13 | 148.61 | 4800 | 0.99 | 34.55 |
| 6652 | Malmsbury | -37.21 | 144.37 | 18 | 0.92 | 43.73 |
| 6655 | Lauriston | -37.27 | 144.39 | 20 | 0.80 | 13.98 |
| 6656 | Upper Coliban | -37.29 | 144.39 | 37.5 | 0.80 | 57.93 |
| 6657 | Rosslynne | -37.47 | 144.57 | 24.5 | 0.93 | 75.76 |
| 6658 | White Swan | -37.52 | 143.92 | 14.1 | 0.91 | 23.12 |
| 6659 | Yan Yean | -37.55 | 145.13 | 32.7 | 0.93 | 43.44 |
| 6662 | Greenvale | -37.63 | 144.90 | 27.5 | 0.83 | 11.88 |
| 6663 | Maroondah | -37.64 | 145.56 | 28.4 | 0.65 | 46.40 |
| 6664 | Upper Yarra | -37.67 | 145.90 | 207.2 | 0.58 | 38.91 |
| 6667 | Silvan | -37.84 | 145.42 | 40.2 | 0.27 | 8.21 |
| 6668 | Glenmaggie | -37.91 | 146.80 | 190 | 0.84 | 39.69 |
| 6669 | Cardinia | -37.97 | 145.39 | 288.9 | 0.90 | 19.06 |
| 6670 | Tarago | -38.02 | 145.94 | 37.5 | 0.78 | 13.47 |
| 6673 | Devilbend | -38.29 | 145.11 | 14.5 | 0.93 | 9.82 |
| 6676 | West Barwon | -38.53 | 143.72 | 21.7 | 0.74 | 61.63 |
| 6701 | Awoonga High | -24.07 | 151.31 | 300 | 0.96 | 40.04 |
| 6702 | Callide | -24.37 | 150.62 | 127 | 0.96 | 49.09 |
| 6703 | Cania | -24.65 | 150.98 | 89 | 0.93 | 60.31 |
| 6704 | Fred Haigh | -24.87 | 151.85 | 586 | 0.97 | 15.20 |
| 6706 | Glebe Weir | -25.46 | 150.03 | 17.3 | 0.69 | 35.41 |
| 6707 | Boondooma | -26.10 | 151.43 | 212 | 0.93 | 11.58 |

**Table 1** The SMAPE and Pearson correlations between predicted and reference volumes for 65 reservoirs (continued).

| Grand ID | Dam Name | Latitude | Longitude | Capacity (MCM) | R | SMAPE (%) |
|---|---|---|---|---|---|---|
| 6708 | Bjelke-Petersen | -26.30 | 151.98 | 125 | 0.98 | 13.87 |
| 6709 | Borumba | -26.51 | 152.58 | 42.6 | 0.91 | 13.56 |
| 6715 | Cressbrook | -27.26 | 152.20 | 83 | 0.98 | 29.38 |
| 6717 | Perseverance Creek | -27.30 | 152.12 | 30.9 | 0.95 | 27.42 |
| 6723 | Moogerah | -28.04 | 152.54 | 92.5 | 0.92 | 45.60 |
| 6725 | Maroon | -28.19 | 152.65 | 38.4 | 0.95 | 21.53 |
| 6726 | Leslie | -28.22 | 151.92 | 108 | 0.98 | 26.93 |
| 6728 | Coolmunda | -28.44 | 151.22 | 75.2 | 0.93 | 18.06 |
| 6731 | Glenlyon | -28.98 | 151.46 | 254 | 0.91 | 23.18 |
| 6731 | Glenlyon | -28.97 | 151.45 | 254 | 0.92 | 20.47 |
| 6733 | Copeton | -29.90 | 150.92 | 1364 | 0.81 | 42.56 |
| 6735 | Split Rock Dam | -30.58 | 150.70 | 372 | 0.95 | 39.52 |
| 6736 | Keepit Dam | -30.88 | 150.49 | 423 | 0.93 | 23.43 |
| 6737 | Chaffey | -31.35 | 151.14 | 61.8 | 0.98 | 12.27 |
| 6738 | Glenbawn | -32.10 | 150.99 | 750 | 0.99 | 9.33 |
| 6739 | Chichester | -32.24 | 151.69 | 17.7 | 0.47 | 15.85 |
| 6740 | Lostock | -32.33 | 151.46 | 20 | 0.63 | 9.90 |
| 6741 | Glennies Creek | -32.36 | 151.25 | 283 | 0.97 | 18.70 |
| 6742 | Grahamstown | -32.77 | 151.79 | 152.6 | 0.84 | 18.31 |
| 6743 | Mangrove Creek | -33.22 | 151.13 | 170 | 0.93 | 49.24 |

**Table 2** The SMAPE and Pearson correlations of predicted volumes between Group A and B for 33 reservoirs.

| Grand ID | Dam Name | Latitude | Longitude | Capacity (MCM) | R | SMAPE (%) |
|---|---|---|---|---|---|---|
| 250 | Mica | 52.08 | -118.57 | 25000 | 0.90 | 15.22 |
| 253 | Gardiner | 51.27 | -106.86 | 9870 | 0.84 | 3.41 |
| 297 | Libby | 48.41 | -115.32 | 7434.2 | 0.89 | 15.97 |
| 310 | Grand Coulee | 47.95 | -118.98 | 6395.6 | 0.92 | 13.19 |
| 370 | Cascade | 44.52 | -116.05 | 805.5 | 0.98 | 15.44 |
| 597 | Glen Canyon | 36.94 | -111.49 | 25070 | 0.99 | 22.43 |
| 1275 | Sam Rayburn Dam And Reservoir | 31.07 | -94.11 | 7815.6 | 0.94 | 5.40 |
| 1320 | International Falcon Lake Dam | 26.56 | -99.17 | 3920 | 0.96 | 13.66 |
| 1863 | Buford | 34.16 | -84.07 | 3150.3 | 0.93 | 9.09 |
| 2376 | Itumbiara | -18.41 | -49.10 | 17000 | 0.96 | 7.81 |
| 2377 | Emborcacao | -18.45 | -47.99 | 17590 | 0.97 | 6.37 |
| 2388 | Mascarenhas de Moraes | -20.28 | -47.06 | 4040 | 0.91 | 3.32 |
| 2405 | Capivara | -22.66 | -51.36 | 10540 | 0.93 | 5.23 |
| 2416 | Paraibuna | -23.36 | -45.66 | 4732 | 0.99 | 5.29 |
| 2447 | Passo Fundo | -27.55 | -52.74 | 1570 | 0.97 | 6.51 |
| 2467 | Araras | -4.21 | -40.45 | 1000 | 0.98 | 12.13 |
| 2490 | Boa Esperanca | -6.75 | -43.57 | 5060 | 0.94 | 9.72 |
| 3014 | Bagre | 11.47 | -0.55 | 1700 | 0.95 | 5.83 |
| 3670 | Mape | 6.04 | 11.30 | 3300 | 0.94 | 8.34 |
| 4212 | Sterkfontein | -28.39 | 29.02 | 2620 | 0.99 | 27.54 |
| 4431 | Karakaya | 38.23 | 39.14 | 9580 | 0.87 | 7.70 |
| 4500 | Nyumba ya Mungu | -3.82 | 37.47 | 1135 | 0.89 | 12.36 |
| 4501 | Mtera | -7.14 | 35.98 | 3200 | 0.98 | 13.74 |
| 4686 | Kayrakkum | 40.28 | 69.82 | 4160 | 0.98 | 4.58 |
| 4702 | Tarbela | 34.09 | 72.69 | 13940 | 0.74 | 29.99 |
| 4715 | Kajakai | 32.32 | 65.12 | 2680 | 0.86 | 12.17 |
| 4739 | Ukai | 21.26 | 73.60 | 8510 | 0.80 | 16.18 |
| 4943 | Upper Indrawati | 19.28 | 82.83 | 2300 | 0.99 | 41.12 |
| 5150 | Lam Pao | 16.60 | 103.45 | 1430 | 0.97 | 9.93 |
| 5796 | Sirindhorn | 15.21 | 105.43 | 1966 | 0.97 | 8.38 |
| 5902 | Shuifeng | 40.46 | 124.97 | 14700 | 0.86 | 13.83 |
| 6606 | Lake Victoria | -34.04 | 141.28 | 680 | 0.98 | 33.63 |
| 6653 | Eildon | -37.22 | 145.93 | 3390 | 0.98 | 16.10 |

[Figure]

**Figure 1** The number of reservoirs with Landsat observations for each month from 1984 - 2015 in the reservoir area product developed by Zhao and Gao (2018).

[Figure]

**Figure 2** The comparison (dash grey line: 1: 1 line) of the linear trends in net evaporation using evaporation losses derived from the W3 model, TerraClimate, NLDA and GLDAS for 721 reservoirs.

---

## Author Comment (AC2)

**Response to Reviewer #2 Comments:**

General Comments

This study demonstrates an integrated remote sensing framework for improving the understanding of long-term reservoir storage dynamics at the global scale. The methods of this study highlight a combination of well-established quantitative approaches and publicly available data sets and have the potential to benefit studies across water resources management and satellite remote sensing. The manuscript is well written and organized, but further explanation or clarification might be needed on the hydrology part, particularly for some components of trend analysis and associated conclusions.

**We thank the reviewer for the thoughtful comments and constructive suggestions, which will help us to improve the quality of the manuscript. Below please find our response to reviewer's comments in detail.**

Specific Comments

R2C1) My major concern is that the trend analysis didn't include reservoir outflow and water use at the reservoir or basin level. The authors did attempt to explain the lack of data behind their decision, but this may not be sufficient to justify an incomplete analysis of the reservoir water balance. Without a reasonable estimation of the dynamics of outflow and water use, it is not convincible that the trend in precipitation/stemflows alone can effectively explain the trend in reservoir storage, particularly for those reservoirs where the trends in precipitation/streamflow and storage are not consistent. Therefore, some of the conclusions on the influence of water use are not robust, e.g., lines 17-18, 221-223, 248-249, 267-268, 362-365, and 376-377.

**We thank the review for this comment. Attributing the causes of reservoir storage change is at the same time important and challenging. There are no water demand and supply or dam operation data available globally (and even very hard to come by locally), and so we are not able to access the influence of human activities on reservoirs directly using such data. Instead, the underlying principle of this study is that the water volume dynamics in a reservoir are the net balance of inflow (streamflow, driven by precipitation), net evaporation (i.e., evaporation minus direct precipitation) and reservoir releases. Based on this, we analysed the individual terms inflow (temporal correlation) and net evaporation (trend ratio in volume) and then, where possible, deduced the role of dam water releases as a residual. This indirect method is the only approach possible given lacking water release data, but by applying logic to the result we were still able to make insightful deductions.**

**Thus, for the majority of the 65 basins with significant storage changes, trends were of the same sign for storage, runoff and precipitation (Fig.7; L269-271). If rainfall and runoff trends show the same directions as reservoir storage, then it is most plausible that climate variations play an important role in reservoir storage trends. On the other hand, if rainfall/runoff and reservoir storage show opposite trends, would that constitute evidence that either direct evaporation or water releases are the driving process, and we were able to exclude the former as a driving**

process. We propose that this logical framework is very robust but welcome arguments as to why it might not be.

There are other recent studies that come to similar conclusion about the limited impact of water releases. For example, Wang et al. (2017) found that climate variability was the dominant driver of the decreasing area trend of lakes across China's Yangtze Plain while human activities only accounted 10-20% of these lake changes, even though the Three Gorges Dam was constructed upstream. Yang et al. (2021) demonstrated that climate variations dominate flood changes in China although there are more dams constructed and land use has changed. We will include such additional evidence in the revised manuscript.

Summarising, we agree with the reviewer that we do not have direct evidence on reservoir releases (or water use) and thereby some of the conclusions on this front are not as robust as we might have liked. Nonetheless, we argue that our interpretation is coherent and logical and still provides insightful evidence. We will however temper the relevant statements to acknowledge the indirect nature of our evidence, for example in L17-18:

> "*Many of the observed reservoir changes were explained well by changes in precipitation and river inflows, emphasising the importance of multi-decadal precipitation changes for reservoir water storage. The results also indicated that there is little impact of changes in net evaporation on storage trends. A more definitive conclusion about any contribution of changes in water releases at global scale would require data that are currently not shared, but we deduce it is unlikely that water release trends dominate global trends.*"

in L248-249:

> "*If precipitation and runoff trends show the same direction as reservoir storage trends, then it is plausible that climate variations play an important role in reservoir storage trends. On the other hand, if rainfall and runoff show opposite trends to those in reservoir storage, then that could suggest evidence of a dominant influence from either net evaporation or water releases. For the majority of these 65 basins, trends were of the same sign for storage, runoff and precipitation, suggesting that precipitation changes are ultimately the most likely explanation for observed trends (Fig. 7a and b).*"

in L362-365:

> "*Both lakes and reservoirs are influenced by changing inflow and net evaporation in response to climate variability. Although human regulation has more influence on reservoirs than on natural lakes, our results suggest that for the majority of basins natural influences dominate human impacts, although these may still exist. For example, Cooley et al. (2021) found that human interventions have resulted in larger seasonal variability in reservoirs than that in lakes globally.*"

in L374-377

> "*Given that reservoir storage dynamics are the net result of river inflows, net evaporation and*

*dam water releases, we found a reasonably strong relationship between changes in river flow and reservoir storage, while changes in net evaporation do not seem to have affected storage trends significantly. We infer that human activities, and specifically reservoir releases, are less likely to be the dominant driver of three-decadal trends in reservoir storage, but acknowledge that our evidence for this conclusion is of an indirect nature, and would require corroboration for an individual reservoir using actual release data"*

**[1] Wang, J., Sheng, Y., & Wada, Y. (2017). Little impact of the Three Gorges Dam on recent decadal lake decline across China's Yangtze Plain. Water Resources Research, 53(5), 3854-3877.**

**[2] Yang, L., Yang, Y., Villarini, G., Li, X., Hu, H., Wang, L., Blöschl, G. & Tian, F. (2021). Climate More Important for Chinese Flood Changes than Reservoirs and Land Use. Geophysical Research Letters, 48(11), e2021GL093061.**

**[3] Cooley, S. W., Ryan, J. C., & Smith, L. C. (2021). Human alteration of global surface water storage variability. Nature, 591(7848), 78-81.**

R2C2) The analysis of reservoir reliability, resilience, and vulnerability (lines 172-189) is a good extension to the estimated reservoir storage dynamics. The concepts and calculations in this part could be better introduced by using a real reservoir as an example, perhaps a well-known reservoir with good data availability. Also, how did the authors determine the time length of failure events (line 178) determined? How does the value of this factor vary among different reservoirs or basins? What is the unit of resilience (line 185)?

**We thank the review for this suggestion. The time length of failure event is defined as the number of continuous months when the storage level drops below 10% lowest value (please see "Duration Time (month)" in Table 3). The time length of failure event is converted to the resilience index using the Eq.6 (L185) from to the previous studies (Hashimoto et al. 1982; Kjeldsen and Rosbjerg 2004). The resilience index ranges from 0 to 1 and has no unit. The lower index it is, the slower recover rate (weakened resilience) the reservoir has, and vice versa. We will include a real reservoir as an example to introduce reliability, resilience, and vulnerability (Fig.3 and Table 3).**

R2C3) Field observations and modeling studies have shown that evaporative loss from reservoir surface can be quite significant, especially for reservoirs in arid and semi-arid regions. This seems to be contradictory to some conclusions from this study (lines 265-266, 307-308 and 311).

**Thank you for this comment. We agree that evaporative losses from reservoirs are large in some arid and semi-arid regions. However, firstly, evaporative losses are relatively more significant in small reservoirs (Mady et al., 2020). Secondly, large evaporative losses affect seasonal storage dynamics but this does not necessarily mean that trends in evaporation explain a long-term trend in reservoir storage. We will include this in the revised manuscript. In addition, we will add the validation of trend analyses of net evaporation against Zhao and Gao (2019), referring to our response to R1C3 for full details (the second paragraph).**

**[4] Mady, B., Lehmann, P., Gorelick, S. M., & Or, D. (2020). Distribution of small seasonal**

**reservoirs in semi-arid regions and associated evaporative losses. Environmental Research Communications, 2(6), 061002.**

Technical Corrections

R2C4) Figures 2-3. No need to use the second y-axis.

**We will change it to use the same vertical scale**

R2C5) Line 171. Remove the comma.

**We will remove the comma in this sentence.**

[Figure]

**Figure 3** Example storage time series showing the definition of resilience and vulnerability (black shade: unsatisfactory state; grey shade: satisfactory state, black line: temporal storages; dash line: 10% threshold; letters: failure events).

**Table 3** The statistics of resilience and vulnerability for the reservoir in Fig. 3.

| Period | 1984-2000 | | | | 2000-2015 | | |
|---|---|---|---|---|---|---|---|
| Failure Event | A | B | C | D | E | F | G |
| Duration Time (month) | 2 | 4 | 3 | 3 | 5 | 3 | 18 |
| Resilience (1/average duration) | | 0.33 | | | | 0.12 | |
| Deficit Volume (GL) | 239 | 589 | 202 | 399 | 329 | 373 | 792 |
| Vulnerability (average deficit volume) | | 357 | | | | 498 | |

---

## Author Comment (AC3)

**Response to Reviewer #3 Comments:**

This study presents a multi-satellite remote sensing approach to understand long term storage changes in over six thousand reservoirs around the world. The authors combine well-established remote sensing based reservoir monitoring techniques to build monthly time series of storage variations. These variations are then synthesized with streamflow to provide insight into long term trends. This is an important study that pushes the boundaries of our understanding of global reservoir storage variations and explores possible drivers of the observed changes. However, I have two major concerns and several minor concerns that should be addressed before publication.

**We would like to thank the reviewer for the thoughtful comments and suggestions. The reviewer provided us with helpful comments, which will greatly improve our manuscript. Below please find our response to reviewer's comments in detail.**

R3C1) First, I am unsure of the value of using long term trends to characterize reservoir storage as increasing or decreasing between 1985 and 2015 (as in lines 210-240). Figure 2 suggests that reservoirs of these sizes can go through shorter, but still multi-year periods of increased and deceased storage throughout this time period. For example, Fort Peck and Fairbairn Reservoirs show ~10 year long oscillations in storage that are not easily characterized by simply increasing or decreasing trends.

**We agree. For our long-term analysis, we not only calculated whether there were increasing or decreasing trends from 1984-2015, but also tested whether these trends were significant or not using the Mann-Kendall trend test ($p<0.05$). The red and blue points in Fig.4 (L225-227) and the basins with black outlines in Fig.7 (L269-271) showed significant increasing or decreasing trends in storage. The changes in storages for Fort Peck and Fairbairn Reservoirs are non-significant trends according to the Mann-Kendall trend test ($p<0.05$). In contrast, for example, the change in storage is significant in one basin of southwestern USA, although there are 10-year-long oscillations in storage (Fig.4).**

R3C2) Second, I am unconvinced of the conclusion that human intervention is an insignificant contribution to storage variability. According to equation 8, changes in storage are related to Qin and Qout (assuming small E). One could argue that any change in storage is due to human alteration of Qout, because without modification of Qout (relative to Qin) there would be no storage variation at all. Without some quantification of the drivers of Qout (hydropower demand, irrigation needs, etc.) I find it hard to make an argument for Qin to be the dominant driver with only what has been quantified in this study. Perhaps an alternate way to frame your findings is that Qin can be used as a good predictor of positive or negative reservoir storage variations.

**We thank the reviewer for this constructive suggestion. We agree that the conclusions on the influence of human intervention are not as robust as desirable without direct evidence, but we would argue that our logic to deduce the role of human activity is reasonable and our conclusions sufficiently circumspect, although we will look for phrasing that reflects that better. We are also able to refer to some individual regional studies that came to a similar conclusion in the revised manuscript. Please refer to our response to R2C1 for full details.**

Line comments:

R3C3) Lines 65-79: The limitations of past efforts and techniques are summarized well here, but how this study overcomes these limitations and provides something new should also be given a sentence or two here.

**Thank you, we will add a few sentences to highlight the advancement of this study over previous ones.**

R3C4) Line 125: This figure could use a legend describing what the colors and inner and outer rings area.

**We will add a legend to this figure**

R3C5) Line 150: Would reservoirs constructed during the study period have an impact on the quantified Qin for older reservoirs?

**Thank you for this question. The hydrological modeling used in this study does not simulate human interventions on river flows and therefore would not reflect any such changes.**

R3C6) Line 171-190: I was confused by the methods for calculating reliability, resilience, and vulnerability. How does assuming 90% reliability simplify the calculations? Why is this a reasonable assumption?

**We apologize for the confusion. We will include a real reservoir as an example to introduce reliability, resilience, and vulnerability. Please see our response to R2C2 for full details.**

R3C7) Line 205: The two vertical axis on Figure 2 and 3 need to be equal for each subplot. As it is now, only correlation is apparent, but it would be much more realistic to plot the observed and predicted on the same vertical scale to get a realistic sense of the errors.

**Thank you for this suggestion. We will change it to use the same vertical scale**

R3C8) Line 342-350: This paragraph felt a little out of place here. Maybe consider moving the content to the methods section.

**Agree. We will move this paragraph to the method section.**

[Figure]

**Figure 4** Total monthly storage dynamics with significant decreasing trend in one basin of southwestern USA.

---

## Author Response (AR1)

**Response to Reviewers**

**''Remotely sensed reservoir water storage dynamics (1984-2015) and the influence of climate variability and management at global scale'' by Jiawei Hou et al.**

**We thank the three reviewers for the thoughtful comments and constructive suggestions, which helped us improve the manuscript. We have thoroughly considered all comments and suggestions, and made modifications accordingly below (review comments in blue, our response in black bold font). The major changes include:**

(1) **We highlighted the advancement of this study over previous ones and clarified the logic of our analysis to determine the extent to which climate variability and human activity each affected global reservoir water volume over the past three decades.**

(2) **We acknowledged the limitation of our approach to deducing the influence of human activity on changes in reservoir storage and the need of in situ records of reservoir water releases to validate this part of our conclusion in specific cases.**

(3) **We implemented higher correlation (R) thresholds between A-L and between A-V for reservoir storage estimation and updated the subsequent long-term analysis.**

(4) **We added a validation of bias and error and included detailed validation results for individual reservoirs globally.**

(5) **We validated trend analyses of net evaporation against Zhao and Gao (2019).**

(6) **We included statistics on the number of reservoirs with Landsat observations for each month from 1984 – 2015.**

(7) **We added a worked example to explain the reliability, resilience, and vulnerability metrics.**

**In addition to replies to reviewers' comments in the open discussion:**

(8) **We included an additional analysis on the influence of human activity on reservoir storage trends using global water withdrawal data.**

**Reviewer #1 Comments:**

In this manuscript, Hou et al. estimated water storage dynamics for more than 6,000 reservoirs worldwide from 1984 to 2015 using a combination of Landsat imagery, radar satellite altimetry, and geostatistical modeling. They also analyzed the patterns of increasing and decreasing trends globally. Finally, they attributed reservoir storage changes to climate and human variables and found that precipitation and river inflows largely dominated reservoir storage changes.

I feel this is a very interesting study. Previous studies provided long-term storage changes for only dozes of reservoirs. It is really great to see a global dataset of more than 6,000 reservoirs, as compiled in this study. Their attributions on the reservoir storage changes can potentially inform local to regional water resources management. However, I have some major concerns on the quality of the global dataset and the methodology that they applied to attribute the storage changes.

R1C1) The Landsat satellites does not provide global coverage in the 1980s and maybe in the 1990s as well (Murray et al 2019). The authors did not acknowledge this limitation while stating they quantified reservoir storage from 1984 to 2015 globally. Is the produced storage time series consistent through 1984 to 2015? Could you provide a figure documenting the number of observations in each year in the time series from 1984 to 2015?

"Murray, N. J. et al. The global distribution and trajectory of tidal flats."

**We agree that Landsat-derived products have limited observations in the 1980s, but this issue predominantly occurs in Oceania, Siberia, Greenland and parts of central and eastern Asia (Pekel et al. (2016); https://www.nature.com/articles/nature20584/figures/5). Landsat-derived water observations are available from 1984 onwards for most parts of Northern America, South America, Africa, Europe, and western and eastern Asia.**

**Furthermore, Zhao and Gao (2018) developed an algorithm to fill gaps in time series when the contamination/occultation in a Landsat image is between 5-95%, and applied interpolation and extrapolation for the missing monthly area estimates (i.e., no images or >95% invalid data). As a result, in their reservoir area product, there are 5,917 reservoirs that have Landsat observations every month from 1984 - 2015 (Fig. R1). To address this point, we changed the sentence in L99-102:**

> "Clouds, cloud shadows and terrain shadows cause errors or missing data for individual months, but Zhao and Gao (2018) developed an automated method to fill gaps in contaminated image classifications and enhance the accuracy and consistency of reservoir surface water extent estimates."

**and modified the sentence in L105-107:**

> "The resulting monthly data are available from 1984 to 2015 and there are 5,917 reservoirs have continuous observations every month over the 32 years. We used this data here as its temporal consistency fits the purpose of this study for long-term trend analysis."

[Figure]

**Figure R1** The number of reservoirs with Landsat observations for each month from 1984 - 2015 in the reservoir area product developed by Zhao and Gao (2018).

**Despite that, we remained careful in using the reservoir area data for long-term storage trend analysis and included additional criteria. First, we removed reservoirs for which more than five years of (not necessarily consecutive) data were inter- or extrapolated. Second, we filtered out reservoirs with observations for less than 360 months (30 years), e.g. in New Zealand. After these steps, we found that 4,573 reservoirs constructed before 1984 have sufficient Landsat observations and these were used for long-term analysis, compared to the 6,669 reservoirs for which we produced monthly storage dynamics.**

**We modified the sentences in L183-186 to explain these steps:**

> *"To ensure consistency in the 1984-2015 time series used for long-term trend analysis, we ignored reservoirs with less than 360 months (i.e., 30 years) of Landsat-derived observations or for which more than five years of water extent observations were inter- or extrapolated by Zhao and Gao (2018)."*

R1C2) While this study produces storage changes for a greater amount of reservoirs globally, I do not think the authors fully addressed the limitations that prevent previous studies from documenting reservoir storage dynamics with a better spatial coverage. The authors estimated storage changes for the 132 large reservoirs with both water areas and levels without assessing their consistency. Without a high correlation between water areas and levels, it makes no sense to me to combine these two to deduce storage changes. The authors need to refer to existing studies (e.g., Busker et al.) on quality control before simply combining satellite observations. The authors used a geostatistical method to estimate the storage changes in the vast majority of reservoirs, on which I have an even greater concern. The authors need to be aware that the mean depth, as archived in the HydroLakes dataset, is a ratio of the total volume and maximum lake area. The mean depth does not provide any meaningful information of the actual water depth. Additionally, the geostatistical model adopted by Messager et al. is a spatial model measuring the relationship between the total storages and maximum areas for a large group of water bodies. The authors tried to use the outcome (e.g., mean depth) to estimate storage changes in each individual reservoir, which differs from the purpose of the Messager et al. Unless the authors provide a comprehensive validation, I am not convinced the proposed method is feasible to estimate storage changes for the majority of studied reservoirs here.

"Busker, T. et al. A global lake and reservoir volume analysis using a surface water dataset and satellite altimetry. Hydrol. Earth Syst. Sci. 23, 669–690 (2019)."

"Messager, M. L., Lehner, B., Grill, G., Nedeva, I. & Schmitt, O. Estimating the volume and age of water stored in global lakes using a geo-statistical approach. Nat. Commun. 7, (2016)."

**We thank the review for this suggestion. Following comments R1C2 and R1C7, we increased correlation (R) thresholds between area and level (A-L) and between area and volume (A-V) for reservoir storage estimation. We regard R values above 0.7 as indicating strong correlation, and used this as the correlation threshold. For group A, we only calculated reservoir storage when the correlation between area and level exceeded 0.7. Storage dynamics between 1984-1993 (when**

**altimetry data is not available) were estimated from area if the correlation with volume exceeded
0.7 between 1993-2015. We changed sentences in L131-134 to explain these steps:**

*"In total, 132 large reservoirs had records of both surface water extent and height for the
overlapping period 1993–2015. Strong correlation (R ≥0.7) between extent and height was found
for 58 reservoirs (Group A; Fig. 1). For these, we estimated the height and area at capacity as the
maximum observed surface water height and extent, respectively, and calculated reservoir storage
volume ($V_o$ in GL or $10^6$ $m^3$) as:"*

**in L137-139:**

*"There were 53 reservoirs with a relationship between $A_o$ and $V_o$ for this overlapping period with
a Pearson's R≥0.7. For these reservoirs, $V_0$ was estimated from 1984 onwards using a cumulative
distribution function (CDF) matching method based on $A_0$."*

**We updated the subsequent long-term analysis as well (from Fig. 1 to Fig. 11; please see in the
revised manuscript). These stricter measures did not in any way affect the conclusions of our
study, but arguably made them more statistically robust.**

**We would like to clarify that we did not directly use the mean depth archived in the
HydroLAKES dataset (Messager et al., 2016). Indeed, this value is not related to the geo-
statistical model and is simply the ratio of the reported volume and lake area. The geo-statistical
model, on the other hand, provides the empirical relationship of the mean depth with water
surface area and the average slope within a 100 m buffer around the water body (Table S2).
Messager et al. (2016) have validated the predicted lake depth and volume derived from the geo-
statistical model against observed data. The symmetric mean absolute percent error (Eq. (R1))
and correlation between predicted and observed lake depth are 47.4% and 0.71, respectively
(Messager et al., 2016). Furthermore, the SMAPE and correlation between predicted and
reference volume are 48.8% and 0.95, respectively (Messager et al., 2016).**

$$SMAPE = 100 \times \frac{1}{N} \sum \frac{\left|\text{observed value} - \text{predicted value}\right|}{\left(\text{observed value} + \text{predicted value}\right)/2} \quad \textbf{(R1)}$$

**We used this statistical model to estimate reservoir depth and volume dynamics from 1984-2015.
We modified the sentences in L154-158 to clarify that the lake depth was predicted by the geo-
statistical model in the revised manuscript:**

*"Messager et al. (2016) proposed a geo-statistical model that provides the empirical relationship
of the mean lake or reservoir depth with water surface area and the average slope within a 100 m
buffer around the water body. The main assumption of this model is that lake bathymetry can be
extrapolated from surrounding topography using slopes. Four empirical equations to predict depth
from area and slope were developed by Messager et al. (2016) for different lake size classes (i.e.,
0.1–1, 1–10, 10–100 and 100–500 $km^2$) (Table S2)."*

**In addition, also responding to comments R1C2 and R1C13, we included the absolute error (SMAPE) in Fig.2 and Fig.3 and listed the SMAPE and correlation metrics for individual reservoirs in supplementary material. We modified the corresponding paragraph in L235-248:**

*"Monthly storage data with at least 20-year time series of 65 reservoirs via the US Army Corps of Engineers and Australian Bureau of Meteorology were used for error assessment. Comparison of observed and estimated volumes showed R>0.9 for 69% of the 65 reservoirs, and R>0.7 for 89% of them (Table S3). Messager et al. (2016) reported that the symmetric mean absolute percent error (SMAPE) of the geo-statistical model is 48.8% globally. The average SMAPE between predicted and reference volumes was 27.8%, lower mainly because we adjusted reservoir storage estimates by reported reservoir capacity. Some cases are shown in Fig. 2. Annual average water levels for Lake Aswan, one of the largest reservoirs in the world, were published as a graph only (El Gammal et al. 2010); comparison showed strong agreement between the satellite-derived storage and in situ measurements (R=0.97, Fig. S1). Assuming the estimation method for Group A is more accurate than that for Group B, the latter can also be evaluated against the former. The results show that 25 of the total 33 overlapping estimated reservoirs show strong agreement (R≥0.9) between the two methods, and the average SMAPE between them is 13.1%. This implies good consistency of reservoir storage estimates from Group A and B. Some cross-validation examples are shown in Fig. 3. The average Pearson correlation between our Landsat-derived water volumes and published MODIS-derived estimates (Tortini et al. 2020; 1992–2015) for 100 reservoirs was R=0.87, and R values did not vary as a function of reservoir size."*

[Figure]

**Figure 2** Validation examples of monthly reservoir storage time series reconstruction against in situ storage data.

[Figure]

**Figure 3** Validation examples of monthly reservoir storage time series reconstruction for Group B against results obtained using the method for Group A.

**Table S3** The SMAPE and Pearson correlation between predicted and reference volumes for 65 reservoirs

| Grand ID | Dam Name | Latitude | Longitude | Capacity (MCM) | R | SMAPE (%) |
|---|---|---|---|---|---|---|
| 307 | Fort Peck Dam | 48.00 | -106.41 | 23560 | 0.98 | 28.6 |
| 597 | Glen Canyon | 36.94 | -111.49 | 25070 | 0.99 | 39.1 |
| 753 | Garrison Dam | 47.51 | -101.43 | 30220 | 0.97 | 31.0 |
| 870 | Oahe Dam | 44.46 | -100.40 | 29110 | 0.97 | 30.5 |
| 6199 | Darwin River Dam | -12.83 | 130.97 | 265 | 0.90 | 8.2 |
| 6579 | Tinaroo Falls | -17.16 | 145.55 | 407 | 0.91 | 11.0 |
| 6581 | Paluma | -18.95 | 146.15 | 12.3 | 0.77 | 18.3 |
| 6582 | Copperfield River Gorge | -19.04 | 144.12 | 20.6 | 0.79 | 14.5 |
| 6583 | Ross River | -19.41 | 146.74 | 417 | 0.95 | 59.5 |
| 6586 | Peter Faust | -20.37 | 148.38 | 500 | 0.94 | 28.7 |
| 6588 | Burdekin Falls | -20.65 | 147.14 | 1860 | 0.89 | 14.7 |
| 6592 | Eungella | -21.14 | 148.39 | 131 | 0.94 | 27.1 |
| 6593 | Kinchant | -21.21 | 148.90 | 62.8 | 0.94 | 22.1 |
| 6594 | Fairbairn | -23.65 | 148.07 | 1440 | 0.96 | 29.6 |
| 6595 | E.J. Beardmore | -27.91 | 148.65 | 101 | 0.84 | 30.5 |
| 6600 | Windamere Dam | -32.73 | 149.77 | 368 | 0.96 | 26.9 |
| 6603 | Carcoar Dam | -33.62 | 149.18 | 35.8 | 0.96 | 12.5 |
| 6605 | Wyangala | -33.97 | 148.95 | 1220 | 0.97 | 22.2 |
| 6613 | Burrinjuck | -35.00 | 148.60 | 1026 | 0.92 | 39.0 |
| 6618 | Blowering | -35.40 | 148.24 | 1628 | 0.92 | 42.0 |
| 6619 | Googong | -35.42 | 149.26 | 124.5 | 0.97 | 7.5 |
| 6620 | Bendora | -35.45 | 148.83 | 11.1 | 0.37 | 12.2 |
| 6621 | Corin | -35.54 | 148.84 | 75 | 0.72 | 20.0 |
| 6629 | Eucumbene | -36.13 | 148.61 | 4800 | 0.99 | 34.6 |
| 6652 | Malmsbury | -37.21 | 144.37 | 18 | 0.92 | 43.7 |
| 6655 | Lauriston | -37.27 | 144.39 | 20 | 0.80 | 14.0 |
| 6656 | Upper Coliban | -37.29 | 144.39 | 37.5 | 0.80 | 57.9 |
| 6657 | Rosslynne | -37.47 | 144.57 | 24.5 | 0.93 | 75.8 |
| 6658 | White Swan | -37.52 | 143.92 | 14.1 | 0.91 | 23.1 |

| 6659 | Yan Yean | -37.55 | 145.13 | 32.7 | 0.93 | 43.4 |
|------|----------|--------|--------|------|------|------|
| 6662 | Greenvale | -37.63 | 144.90 | 27.5 | 0.83 | 11.9 |
| 6663 | Maroondah | -37.64 | 145.56 | 28.4 | 0.65 | 46.4 |
| 6664 | Upper Yarra | -37.67 | 145.90 | 207.2 | 0.58 | 38.9 |
| 6667 | Silvan | -37.84 | 145.42 | 40.2 | 0.27 | 8.2 |
| 6668 | Glenmaggie | -37.91 | 146.80 | 190 | 0.84 | 39.7 |
| 6669 | Cardinia | -37.97 | 145.39 | 288.9 | 0.90 | 19.1 |
| 6670 | Tarago | -38.02 | 145.94 | 37.5 | 0.78 | 13.5 |
| 6673 | Devilbend | -38.29 | 145.11 | 14.5 | 0.93 | 9.8 |
| 6676 | West Barwon | -38.53 | 143.72 | 21.7 | 0.74 | 61.6 |
| 6701 | Awoonga High | -24.07 | 151.31 | 300 | 0.96 | 40.0 |
| 6702 | Callide | -24.37 | 150.62 | 127 | 0.96 | 49.1 |
| 6703 | Cania | -24.65 | 150.98 | 89 | 0.93 | 60.3 |
| 6704 | Fred Haigh | -24.87 | 151.85 | 586 | 0.97 | 15.2 |
| 6706 | Glebe Weir | -25.46 | 150.03 | 17.3 | 0.69 | 35.4 |
| 6707 | Boondooma | -26.10 | 151.43 | 212 | 0.93 | 11.6 |
| 6708 | Bjelke-Petersen | -26.30 | 151.98 | 125 | 0.98 | 13.9 |
| 6709 | Borumba | -26.51 | 152.58 | 42.6 | 0.91 | 13.6 |
| 6715 | Cressbrook | -27.26 | 152.20 | 83 | 0.98 | 29.4 |
| 6717 | Perseverance Creek | -27.30 | 152.12 | 30.9 | 0.95 | 27.4 |
| 6723 | Moogerah | -28.04 | 152.54 | 92.5 | 0.92 | 45.6 |
| 6725 | Maroon | -28.19 | 152.65 | 38.4 | 0.95 | 21.5 |
| 6726 | Leslie | -28.22 | 151.92 | 108 | 0.98 | 26.9 |
| 6728 | Coolmunda | -28.44 | 151.22 | 75.2 | 0.93 | 18.1 |
| 6731 | Glenlyon | -28.98 | 151.46 | 254 | 0.91 | 23.2 |
| 6731 | Glenlyon | -28.97 | 151.45 | 254 | 0.92 | 20.5 |
| 6733 | Copeton | -29.90 | 150.92 | 1364 | 0.81 | 42.6 |
| 6735 | Split Rock Dam | -30.58 | 150.70 | 372 | 0.95 | 39.5 |
| 6736 | Keepit Dam | -30.88 | 150.49 | 423 | 0.93 | 23.4 |
| 6737 | Chaffey | -31.35 | 151.14 | 61.8 | 0.98 | 12.3 |
| 6738 | Glenbawn | -32.10 | 150.99 | 750 | 0.99 | 9.3 |
| 6739 | Chichester | -32.24 | 151.69 | 17.7 | 0.47 | 15.8 |
| 6740 | Lostock | -32.33 | 151.46 | 20 | 0.63 | 9.9 |
| 6741 | Glennies Creek | -32.36 | 151.25 | 283 | 0.97 | 18.7 |
| 6742 | Grahamstown | -32.77 | 151.79 | 152.6 | 0.84 | 18.3 |
| 6743 | Mangrove Creek | -33.22 | 151.13 | 170 | 0.93 | 49.2 |

**Table S4** The SMAPE and Pearson correlation of predicted volumes between Group A and B for 33 reservoirs

| Grand ID | Dam Name | Latitude | Longitude | Capacity (MCM) | R | SMAPE (%) |
|---|---|---|---|---|---|---|
| 250 | Mica | 52.08 | -118.57 | 25000 | 0.90 | 15.2 |
| 253 | Gardiner | 51.27 | -106.86 | 9870 | 0.84 | 3.4 |
| 297 | Libby | 48.41 | -115.32 | 7434.2 | 0.89 | 16.0 |
| 310 | Grand Coulee | 47.95 | -118.98 | 6395.6 | 0.92 | 13.2 |
| 370 | Cascade | 44.52 | -116.05 | 805.5 | 0.98 | 15.4 |
| 597 | Glen Canyon | 36.94 | -111.49 | 25070 | 0.99 | 22.4 |
| 1275 | Sam Rayburn Dam And Reservoir | 31.07 | -94.11 | 7815.6 | 0.94 | 5.4 |
| 1320 | International Falcon Lake Dam | 26.56 | -99.17 | 3920 | 0.96 | 13.7 |
| 1863 | Buford | 34.16 | -84.07 | 3150.3 | 0.93 | 9.1 |
| 2376 | Itumbiara | -18.41 | -49.10 | 17000 | 0.96 | 7.8 |
| 2377 | Emborcacao | -18.45 | -47.99 | 17590 | 0.97 | 6.4 |
| 2388 | Mascarenhas de Moraes | -20.28 | -47.06 | 4040 | 0.91 | 3.3 |
| 2405 | Capivara | -22.66 | -51.36 | 10540 | 0.93 | 5.2 |
| 2416 | Paraibuna | -23.36 | -45.66 | 4732 | 0.99 | 5.3 |
| 2447 | Passo Fundo | -27.55 | -52.74 | 1570 | 0.97 | 6.5 |
| 2467 | Araras | -4.21 | -40.45 | 1000 | 0.98 | 12.1 |
| 2490 | Boa Esperanca | -6.75 | -43.57 | 5060 | 0.94 | 9.7 |
| 3014 | Bagre | 11.47 | -0.55 | 1700 | 0.95 | 5.8 |
| 3670 | Mape | 6.04 | 11.30 | 3300 | 0.94 | 8.3 |
| 4212 | Sterkfontein | -28.39 | 29.02 | 2620 | 0.99 | 27.5 |
| 4431 | Karakaya | 38.23 | 39.14 | 9580 | 0.87 | 7.7 |
| 4500 | Nyumba ya Mungu | -3.82 | 37.47 | 1135 | 0.89 | 12.4 |
| 4501 | Mtera | -7.14 | 35.98 | 3200 | 0.98 | 13.7 |
| 4686 | Kayrakkum | 40.28 | 69.82 | 4160 | 0.98 | 4.6 |
| 4702 | Tarbela | 34.09 | 72.69 | 13940 | 0.74 | 30.0 |
| 4715 | Kajakai | 32.32 | 65.12 | 2680 | 0.86 | 12.2 |
| 4739 | Ukai | 21.26 | 73.60 | 8510 | 0.80 | 16.2 |
| 4943 | Upper Indrawati | 19.28 | 82.83 | 2300 | 0.99 | 41.1 |
| 5150 | Lam Pao | 16.60 | 103.45 | 1430 | 0.97 | 9.9 |
| 5796 | Sirindhorn | 15.21 | 105.43 | 1966 | 0.97 | 8.4 |
| 5902 | Shuifeng | 40.46 | 124.97 | 14700 | 0.86 | 13.8 |
| 6606 | Lake Victoria | -34.04 | 141.28 | 680 | 0.98 | 33.6 |
| 6653 | Eildon | -37.22 | 145.93 | 3390 | 0.98 | 16.1 |

R1C3) The presented attribution on reservoir storage changes seems to be so simplified that I have many concerns. First, the authors simply compared the directions of the trend in reservoir storage versus that in potential drivers but the analysis only produces coincidence rather than causation. Second, the authors conclude that the evaporation did not significantly impact the reservoir storage but the calculation for the evaporation is too cheap. The authors may need to use more advanced approaches (e.g., Zhao and Gao) in order to draw a confident conclusion. Third, reservoirs, particularly

large ones as documented in GranD dataset, are highly regulated by humans. The authors depend on the outputs of global models on estimating human water release from reservoirs. Are the data really reliable for producing trend in human release for each reservoir? In sum, the authors need to pay careful attention to these limitations that potentially affect their conclusions.

"Zhao, G. & Gao, H. Estimating reservoir evaporation losses for the United States: Fusing remote sensing and modeling approaches. Remote Sens. Environ. 226, 109–124 (2019)"

**We thank the reviewer for this comment, but in fact we explicitly considered the difference between coincidence and causation in our study to the extent possible. In a first step, we indeed looked at the coincidence of trends per se. We identified that the spatial distribution of trends of storage and in situ river flow show very similar global patterns (Fig. 4). We could not relate each individual reservoir to a corresponding river gauge because the limited number of gauging stations upstream of reservoirs globally. Instead, in a second step, we performed trend analysis using modeled river flow (validated against in situ river flow in Fig.8b) at the basin scale, given total basin water storage can be expected to respond to a change in overall precipitation and streamflow. We confirmed the same directions of trends between precipitation, streamflow and reservoir storage in most basins, though not all (Fig. 7). Third, we focused more on attribution by calculating Pearson correlations among the different variables, which provides evidence if not proof for a causative relationship. Thus, we showed that there are reasonably strong correlations among linear trends in precipitation, streamflow and reservoir storage (Fig. 8a and c). Furthermore, positive relationships between annual time series of storage change and reservoir inflow and between reservoir inflow and precipitation were found in a majority of basins globally (Fig. 9). We modified the last paragraph (L79-94) in the Introduction section to clarify the logic of our analysis as early as possible in the revised manuscript.**

> *"In this study, we combined Landsat-derived surface water extents, satellite altimetry, and geo-statistical models to reconstruct monthly reservoir storage globally for 1984-2015, and examined long-term trends of global reservoir water storage and changes in reservoir resilience and vulnerability over the past three decades. Part of our objective was to determine the extent to which climate variability and human activity each affected global reservoir dynamics over the past three decades. It is currently impossible to analyse the influence of human activity at global scale directly: there are very few in situ reservoir water release records available publicly, and no hydrological models that can provide reliable estimates. Instead, we consider all climate terms in the reservoir water balance and infer the influence of the remaining unknown term, water releases. First, we investigated trends in precipitation, streamflow and storage at both the reservoir and basin level. If the trends between these variables show similar spatial patterns globally, then this increases the likelihood that climate variability commonly explains storage changes. Second, we examined the temporal correlation between precipitation, reservoir inflow and storage change to further understand potential causative relationships. Third, beyond reservoir releases, net evaporation is the only other potential loss term, and we examined what fraction of observed*

*trends in storage was attributable to net evaporation. Using the combined insights, we deduced the role of human activity on reservoir storage change, noting that a direct attribution would require in situ records of reservoir water releases. To support our inference, we analysed the trends of global water withdrawal to discuss whether it could be a significant factor to lead reservoir storage change."*

**With regards to reservoir evaporation, our estimates are robust. Various hydrological variables estimated by the W3 model have been evaluated in previous studies. Therefore, we argue that the $E_0$ derived from the W3 model are entirely appropriate to analyze linear trends in net evaporation. However, we are also able to provide more direct evidence. Zhao and Gao (2019) estimated evaporation losses for 721 reservoirs in the contiguous United States using three different meteorological datasets, including TerraClimate, North American Land Data Assimilation System phase 2 (NLDAS-2) forcing and Global Land Data Assimilation System Version 2 and Version 2.1 (GLDAS-2 and GLDAS-2.1). We used their monthly reservoir evaporation (1000 $m^3$/month) estimates to compare trends in net evaporation with those in the W3 model estimates we used (Van Dijk et al. 2018) for 721 reservoirs. The results show strong agreement in derived linear trends, especially with regards to the more detailed TerraClimate dataset, which would be expected the most accurate among the three (Fig.R2).**

[Figure]

**Figure R2** The comparison (dash grey line: 1: 1 line) of the linear trends (%) in net evaporation using evaporation losses derived from the W3 model, TerraClimate, NLDA and GLDAS for 721 reservoirs.

**No eartH2Observe models includes the impacts of reservoirs on river flows, and there is currently no global hydrological model that is capable of estimating the impact of historical operational water management at the reservoir or basin level with any acceptable degree of accuracy – hence the need for our approach. We focused on all climate terms of the reservoir water balance and inferred the influence of the remaining unknown variable (i.e. reservoir releases). Specifically, we analysed the interaction between precipitation, streamflow, evaporation and reservoir volume and inferred the influence of human activity given that the water volume dynamics in a reservoir is the net balance of inflow (streamflow, affected by precipitation), net evaporation (i.e., evaporation minus direct precipitation) and reservoir releases (L349-363). We modified the last paragraph (L79-94) in the Introduction section to better explain the logic of our approach to infer the role of human activity in reservoir storage**

**long-term changes early on in the revised manuscript. (Please see the revised text in the first paragraph)**

Specific comments:

R1C4) Line 25: "The majority of …particularly in South America, Southeast Asia and Africa". The authors may consider add more relevant references here.

"Wang, J. et al. GeoDAR: Georeferenced global dam and reservoir database for bridging attributes and geolocations. Earth Syst. Sci. Data 0–52 (2021)"

"Mulligan, M., van Soesbergen, A. & Sáenz, L. GOODD, a global dataset of more than 38,000 georeferenced dams. Sci. Data 7, 1–8 (2020)."

**Thank you. We included these two references in L33-34 in the revised manuscript:**

> *"The majority of these are in developing countries, particularly in South America, Southeast Asia and Africa (Bonnema et al. 2016; Mulligan et al. 2020; Wang et al. 2021; Zarfl et al. 2015)."*

R1C5) Line 65: Schwatke et al. 2019 is another study on estimating long-term lake area changes.

Schwatke, C., Scherer, D. & Dettmering, D. Automated Extraction of Consistent Time-Variable Water Surfaces of Lakes and Reservoirs Based on Landsat and Sentinel-2. Remote Sens. 11, 1010 (2019)

**We included this paper in L69-71 in the revised manuscript:**

> *"Several regional and global time series of reservoir water extent have been produced based on MODIS, Landsat or Sentinel-2 imagery (Khandelwal et al. 2017; Ogilvie et al. 2018; Schwatke et al. 2019; Yao et al. 2019; Zhao and Gao 2018)."*

R1C6) Line 89: It is hard to understand "coefficient of determination" here. Could you define or explain it?

**We used Pearson correlation throughout this paper. We converted coefficient of determination ($R^2$) to Pearson correlation (R) in L104-105 in the revised manuscript:**

> *"The average Pearson correlation (R) between satellite-derived extent and observed elevation or volumes increased from 0.66 to 0.92 using the algorithm developed by Zhao and Gao (2018)."*

R1C7) Line 120: I do not quite understand what's the purpose showing the correlation between A0 and calculated V0 (based on A0). It makes more sense to me to show the correlation between A0 and h0 as these two are independent estimates. The authors may only need to consider a pearson' R greater than 0.8 (or R2 higher than 0.6) as correlation between A-L or A-V should be pretty high, otherwise it indicates substantial uncertainty in the data sets.

**Thank you for this suggestion. We increased correlation (R) thresholds between A-L and between A-V for reservoir storage estimation. Please see our response to R1C2 for full details.**

R1C8) Line 135: the equation does not make sense to me. The authors need to show more details about the rationale.

**We apologize for the confusion. We modified the sentences in L160-166 to clarify the rationale of this equation:**

*"In line with Eq. (1), we assumed maximum observed surface water extent ($A_{max}$) as the area at capacity. Water depth ($D_c$ in m) at capacity was calculated as the ratio of volume ($V_c$) and area ($A_c=A_{max}$) at capacity:*

$$D_c = \frac{V_c}{A_c} \qquad (2)$$

*A bias-corrected water depth ($D^*$ in m) was calculated by solving D based on the ratio of water depth ($D_c$ in m) at capacity and maximum observed depth ($D_{max}$ in m):*

$$D^* = D \times \frac{D_c}{D_{max}} \qquad (3)"$$

R1C9) Line 150: "Only 132 reservoirs with both area and level observations….". Do you conclude based on the 132 reservoirs or all reservoirs, majority of which do not have both observations?

**We performed a long-term trend analysis for 4,573 reservoirs, including both Group A (area and level observations available) and Group B (only area) (L189-190). This sentence aimed to introduce how many reservoirs and how much capacity were estimated in Group A and B. We modified the sentences in L179-183 to clarify this information:**

*"There were only 58 reservoirs for which storage dynamics could be estimated most directly, by a combination of satellite extent and water level observations (Group A), but together they already account for 25.5% of combined global reservoir capacity (Fig. 1). The total capacity of the 193 reservoirs not measured constitutes 36.4% of global capacity. There were 6,611 reservoirs in Group B for which by the geo-statistical approach could be applied, and these contribute 41.1% to total global capacity."*

R1C10) 164: It seems the MSWEP v1.1 may not be the latest version of the dataset.

**Although there is a latest version of MSWEP now, we carried out our study using MSWEP 1.1 two years ago. We considered updating the results with the latest MSWEP product, but among us are authors of the MSWEP product and with knowledge of the changes between successive versions we are confident there are no meaningful differences for the type of long-term analysis done here.**

R1C11) 192: The authors only validated on 1% of the studied reservoirs and the validation samples are located in U.S. only, which could be a concern.

**Please see our response to R1C2 for full details: we provide validation of our calculated storage volumes for 65 reservoirs with publicly available storage data from US and Australia and cross-**

validation between two volume estimation methods for 33 reservoirs globally. We note that the availability of *in situ* data is severely limited, which was the primary reason for us to develop a remote sensing-based methodology. However, to further evaluate our storage estimates, we also compared our estimates with MODIS-derived water storage dynamics for 1992–2018 for another 100 reservoirs by Tortini et al. (2020) (L246-248).

R1C12) 194: What do you mean by "published"? The authors use pearson's R (correlation) for doing validation, which does not give insights on the accuracy of estimated values.

We changed "published" to "observed". We think Pearson correlation is the more relevant metric for this study as we focus on trend analysis, which depends on temporal patterns rather than absolute values. However, we included the absolute error (SMAPE) valuation and compared this validation to Messager et al. (2016) in the revised manuscript. Please see our response to R1C2 for full details.

R1C13) Figure 2: it would be more clear to show global-scale evaluation and move the evaluations on individual cases into the supplementary.

Thank you for this suggestion. We added the validation results for individual reservoirs in the supplementary material. Please see our response to R1C2 for full details.

R1C14) Line 214: "a positive trend in combined global reservoir storage of 3.1 km3 per yr". This rate seems to less than 10% of earlier estimates on global reservoir storage rates (e.g., Chao et al.). Could you provide an uncertainty estimate for this rate?

"Chao, B. F., Wu, Y. H. & Li, Y. S. Impact of artificial reservoir water impoundment on global sea level. Science (80-. ). 320, 212–214 (2008)."

We explicitly performed trend analysis for the reservoirs constructed before 1984 only, in order to remove the influence of new reservoir water impoundments after that year. This was done deliberately, to provide a clearer understanding on the interaction between precipitation, streamflow, evaporation and reservoir volume. Our study therefore differs in key aspects from Chao et al. (2008), who focused on the change in cumulative storage by increased water impoundment. We clarified this information in L186-188 in the revised manuscript:

> *"Our focus was on interactions between precipitation, streamflow, evaporation and storage in existing reservoirs, rather than the consequences of new impoundments. Therefore, we excluded from consideration all reservoirs that were destroyed, modified, planned, replaced, removed, subsumed or constructed after 1984."*

R1C15) Line 215: "this was almost entirely explained by positive trends for the two largest reservoirs in the world, Lake Kariba (+0.8 km3 yr-1) on the Zambezi River and Lake Aswan". This statement is confusing. I know some completed projections of megadams in China and Brazil, such as the three gorges dams.

**Please refer to R1C14. Most mega-dams in China and Brazil were constructed after 1984 and hence were excluded from our long-term analysis.**

R1C16) Line 219: "while 948 reservoirs showed increasing trends, distributed in northern North America and southern Africa". The reported hotspots of increasing reservoir storage are inconsistent with the patterns of recent dam booms.

**Please refer to R1C14.**

R1C17) Figure 4: This map is confusing to me. For example, China may be the global lead in dam constructions during the study period. Why its reservoir storage decreased? Is the data correctly shown in this map?

**Please refer to R1C14.**

R1C18) Line 245: "We summed storage for individual reservoirs to calculate combined storage in 134 river basins worldwide". Do reservoirs show a similar pattern of storage change in the same river basin? Is it more meaningful to analyze each of them individually?

**The majority of reservoirs showed the same trends in each basin where a significant trend in total storage was found (Fig. 11). We show the trend analysis at both the individual (Fig.4a) and basin scale (Fig.7c). Due to the lack of accurate data on the contributing area for each individual reservoir, we performed our climate analysis at the basin scale. The strong spatial correlation in all variables involved meant that combined basin reservoir water storage can be related to changes in basin-average precipitation and streamflow. However, we do find and explicitly discuss cases where those 'average' relationships appeared to break down.**

R1C19) Line 268: "In summary, we did not find evidence for widespread reductions in reservoir water storage due to increased releases". Reservoir storage increase could be a result of increased impoundments. Did you consider that?

**Please refer R1C14. We have removed the effect of the increased reservoir water impoundments from 1984-2015. Therefore, reservoir storage increase cannot be a result of increased impoundments.**

R1C20) Line 339: As Zhao and Gao used contaminated Landsat imagery to increase the monthly coverage of reservoir areas by 80%, do the estimates from poor-quality images affect your storage analysis? I know some studies (e.g., Busker et al) only adopted good-quality images due to this issue.

"Busker, T. et al. A global lake and reservoir volume analysis using a surface water dataset and satellite altimetry. Hydrol. Earth Syst. Sci. 23, 669–690 (2019)."

**We think the consistency of observations from 1984-2015 is more important for long-term analysis. This is the reason why we used the reservoir area product developed by Zhao and Gao (2018). In addition, Zhao and Gao (2018) demonstrated that the correlations between observed and estimated reservoir areas were improved from 0.66 to 0.92 by 'repairing' contaminated Landsat images.**

**However, we did include a reference to the study of Busker et al., as it is certainly relevant to the topic.**

Reviewer #2 Comments:

General Comments

This study demonstrates an integrated remote sensing framework for improving the understanding of long-term reservoir storage dynamics at the global scale. The methods of this study highlight a combination of well-established quantitative approaches and publicly available data sets and have the potential to benefit studies across water resources management and satellite remote sensing. The manuscript is well written and organized, but further explanation or clarification might be needed on the hydrology part, particularly for some components of trend analysis and associated conclusions.

Specific Comments

R2C1) My major concern is that the trend analysis didn't include reservoir outflow and water use at the reservoir or basin level. The authors did attempt to explain the lack of data behind their decision, but this may not be sufficient to justify an incomplete analysis of the reservoir water balance. Without a reasonable estimation of the dynamics of outflow and water use, it is not convincible that the trend in precipitation/stemflows alone can effectively explain the trend in reservoir storage, particularly for those reservoirs where the trends in precipitation/streamflow and storage are not consistent. Therefore, some of the conclusions on the influence of water use are not robust, e.g., lines 17-18, 221-223, 248-249, 267-268, 362-365, and 376-377.

**We thank the review for this comment. Attributing the causes of reservoir storage change is at the same time important and challenging. There are no water demand and supply or dam operation data available globally (and even very hard to come by locally), and so we are not able to assess the influence of human activities on reservoirs directly using such data. The underlying principle of our study is that the water volume dynamics in a reservoir are the result of a relatively simple water balance involving inflow (streamflow, driven by precipitation), net evaporation (i.e., evaporation from minus direct precipitation onto the reservoir) and reservoir releases. Following that logic, we analysed the individual terms inflow (temporal correlation) and net evaporation (trend ratio in volume) and then, where possible, deduced the role of dam water releases as the residual. This indirect method was the only approach open to us, given the lack of water release data, but thanks to the small number of terms in the reservoir water balance, applying this logic we were still able to derive useful insights.**

**Thus, for the majority of the 65 basins with significant storage changes, trends were of the same sign for storage, runoff and precipitation (Fig.7). If rainfall and runoff trends show the same directions as reservoir storage, then it is plausible that rainfall variations dominate reservoir storage trends. On the other hand, if rainfall/runoff and reservoir storage show *opposite* trends, that strongly suggests that direct evaporation or water releases (or both) are the driving process.**

We were able to exclude the former as a driving process. We propose that this logical framework is robust but welcome arguments as to why it might not be.

Incidentally, there are other recent studies that come to similar conclusions about the limited impact of water releases. We included these studies to support our findings in L359-363 in the revised manuscript:

> *"Evidence that the impact of human activity is less than that of climate variability is also found in other recent studies. For example, Wang et al. (2017) found that climate variability was the dominant driver of the decreasing trend in lake area across China's Yangtze Plain; human activities only accounted 10-20% of trends despite construction of the Three Gorges Dam upstream. Furthermore, Gudmundsson et al. (2021) demonstrated that climate change dominates changes in river flow from 1971-2010 worldwide, rather than water and land management."*

Overall, however, we agree with the reviewer that we do not have direct evidence on reservoir releases (or downstream water use) and that, although our interpretation is coherent and logical, some of the associated conclusions are not as robust as we might have preferred. Hence, on one hand, we tempered the relevant statements to acknowledge the indirect nature of our evidence, for example in L18-24:

[revised manuscript text omitted]

R2C2) The analysis of reservoir reliability, resilience, and vulnerability (lines 172-189) is a good extension to the estimated reservoir storage dynamics. The concepts and calculations in this part could

be better introduced by using a real reservoir as an example, perhaps a well-known reservoir with good data availability. Also, how did the authors determine the time length of failure events (line 178) determined? How does the value of this factor vary among different reservoirs or basins? What is the unit of resilience (line 185)?

**We thank the reviewer for this suggestion. The time length of failure event is defined as the number of continuous months when the storage level drops below 10% lowest value (see "Duration Time (month)" in Table 3). The time length of failure event is converted to the resilience index using Eq.6, following previous studies (Hashimoto et al. 1982; Kjeldsen and Rosbjerg 2004). The resilience index has a unit of month$^{-1}$ and ranges from 0 to 1 in this study. A lower index value indicates a slower recovery rate (weakened resilience). We included a worked example for an actual reservoir to introduce reliability, resilience, and vulnerability (Fig. S5 and Table S5 in the supplementary material). We also added some text to describe this example in L225-229 in the revised manuscript:**

*"A worked example is shown for the Toledo Bend Reservoir (Texas, USA) (Fig. S5 and Table S5). Four failure events occurred during 1984–2000 and three during 2000–2015. Before 2000, it took an average of three months to recover from failure, with an average deficit volume of 357 GL. After 2000, it took an average of 10.5 months with a larger average deficit volume of 498 GL (Fig. S1). It follows that resilience was reduced (resilience index 0.12 vs. 0.33) and vulnerability increased (deficit volume 498 vs. 357 GL) when compared to the years before 2000 (Table S5)."*

[Figure]

**Figure S5** Example storage time series showing the definition of resilience and vulnerability (black shaded: unsatisfactory state; grey shaded: satisfactory state, dashed line: 10% threshold; letters: labeled failure events).

**Table S5** The statistics of resilience and vulnerability for the reservoir in Fig. S5.

| Period | 1984-2000 | | | | 2000-2015 | | |
|---|---|---|---|---|---|---|---|
| Failure Event | A | B | C | D | E | F | G |
| Duration Time (month) | 2 | 4 | 3 | 3 | 5 | 3 | 18 |
| Resilience (1/month) | | 0.33 | | | | 0.12 | |
| Deficit Volume (GL) | 239 | 589 | 202 | 399 | 329 | 373 | 792 |
| Vulnerability (average deficit volume) | | 357 | | | | 498 | |

R2C3) Field observations and modeling studies have shown that evaporative loss from reservoir surface can be quite significant, especially for reservoirs in arid and semi-arid regions. This seems to be contradictory to some conclusions from this study (lines 265-266, 307-308 and 311).

**Thank you for this comment. We also found evidence that evaporative losses from reservoirs are large in arid and semi-arid regions, but they did not explain long-term trends. Large evaporative losses tend to affect seasonal storage dynamics but this does not necessarily mean that *trends* in evaporation explain a long-term *trend* in reservoir storage. As an aside, evaporative losses also tend to be less significant in large reservoirs due to their greater depth (Mady et al., 2020).**

**We included this statement in L351-354 in the revised manuscript:**

*"Mady et al. (2020) and various other authors found that evaporative losses can account for much of the loss of water from small reservoirs (e.g., <0.1 km$^2$) in semi-arid regions. However, this does not necessarily mean that trends in evaporation can explain trends long-term trends in storage, especially for the mostly larger (and deeper) reservoirs considered here."*

**In addition, we show the validations of trend analyses of net evaporation against Zhao and Gao (2019), referring to our response to R1C3 for full details.**

**[6] Mady, B., Lehmann, P., Gorelick, S. M., & Or, D. (2020). Distribution of small seasonal reservoirs in semi-arid regions and associated evaporative losses. Environmental Research Communications, 2(6), 061002.**

Technical Corrections

R2C4) Figures 2-3. No need to use the second y-axis.

**We changed it to use the same vertical scale.**

R2C5) Line 171. Remove the comma.

**We removed the comma in this sentence.**

**Reviewer #3 Comments:**

This study presents a multi-satellite remote sensing approach to understand long term storage changes in over six thousand reservoirs around the world. The authors combine well-established remote sensing based reservoir monitoring techniques to build monthly time series of storage variations. These variations are then synthesized with streamflow to provide insight into long term trends. This is an important study that pushes the boundaries of our understanding of global reservoir storage variations and explores possible drivers of the observed changes. However, I have two major concerns and several minor concerns that should be addressed before publication.

R3C1) First, I am unsure of the value of using long term trends to characterize reservoir storage as increasing or decreasing between 1985 and 2015 (as in lines 210-240). Figure 2 suggests that reservoirs

of these sizes can go through shorter, but still multi-year periods of increased and deceased storage throughout this time period. For example, Fort Peck and Fairbairn Reservoirs show ~10 year long oscillations in storage that are not easily characterized by simply increasing or decreasing trends.

**We agree. For our long-term analysis, we not only calculated whether there were increasing or decreasing trends from 1984-2015, but also tested whether these trends were significant or not, using the Mann-Kendall trend test (p<0.05). The red and blue points in Fig.4 and the basins with black outlines in Fig.7 showed significant increasing or decreasing trends in storage. The Fort Peck and Fairbairn Reservoirs show non-significant trends according to the Mann-Kendall trend test. In contrast, for example, the change in total storage is significant in Colorado River Basin (including 76 reservoirs) of southwestern USA, although there are decadal oscillations in storage (Fig.R4).**

[Figure]

**Figure R4** Total monthly storage dynamics with significant decreasing trend in one basin of southwestern USA.

R3C2) Second, I am unconvinced of the conclusion that human intervention is an insignificant contribution to storage variability. According to equation 8, changes in storage are related to Qin and Qout (assuming small E). One could argue that any change in storage is due to human alteration of Qout, because without modification of Qout (relative to Qin) there would be no storage variation at all. Without some quantification of the drivers of Qout (hydropower demand, irrigation needs, etc.) I find it hard to make an argument for Qin to be the dominant driver with only what has been quantified in this study. Perhaps an alternate way to frame your findings is that Qin can be used as a good predictor of positive or negative reservoir storage variations.

**We thank the reviewer for this constructive suggestion. We agree that the conclusions on the influence of human intervention are not as robust as we might like, given the lack of publicly available records on releases. However, we believe our logic to deduce the role of human activity is sound and our conclusions sufficiently cautious, especially in the revised manuscript. Besides, we did further analysis on human impacts using global water withdrawal estimates (Huang et al., 2018) as an approximation of water releases. This analysis provides additional evidence on the conclusion that climate trends rather than water withdrawals are primarily responsible for the observed trends in reservoir storage. Please refer to our response to R2C1 for full details.**

Line comments:

R3C3) Lines 65-79: The limitations of past efforts and techniques are summarized well here, but how this study overcomes these limitations and provides something new should also be given a sentence or two here.

**Thank you, we modified the last paragraph (L79-94) in the introduction to highlight the advancement of this study over previous ones. Please refer to our response to R1C3 for full details.**

R3C4) Line 125: This figure could use a legend describing what the colors and inner and outer rings area.

**We added a legend to this figure.**

[Figure]

**Figure 1** The total storage capacity in Group A (red) and B (brown) and left unaccounted (blue) and the combined capacity of reservoirs for which the data were suitable (teal) or unsuitable (pink) for long-term analysis.

R3C5) Line 150: Would reservoirs constructed during the study period have an impact on the quantified Qin for older reservoirs?

**Thank you for this question. The hydrological models do not simulate the effect of reservoirs and river operations on flows and therefore would not reflect any such changes.**

R3C6) Line 171-190: I was confused by the methods for calculating reliability, resilience, and vulnerability. How does assuming 90% reliability simplify the calculations? Why is this a reasonable assumption?

**We apologize for the confusion. We included an actual reservoir as an example to introduce reliability, resilience, and vulnerability. Please see our response to R2C2 for full details.**

R3C7) Line 205: The two vertical axis on Figure 2 and 3 need to be equal for each subplot. As it is now, only correlation is apparent, but it would be much more realistic to plot the observed and predicted on the same vertical scale to get a realistic sense of the errors.

**Thank you for this suggestion. We changed it to use the same vertical scale in Fig. 2 and 3.**

R3C8) Line 342-350: This paragraph felt a little out of place here. Maybe consider moving the content to the methods section.

**Agree. We moved this paragraph to the method section in L144-150 in the revised manuscript.**

---

## Author Response (AR2)

**''Remotely sensed reservoir water storage dynamics (1984-2015) and the influence of climate variability and management at global scale'' by Jiawei Hou et al.**

**We thank the editor and the second reviewer for the thoughtful comments and constructive suggestions, which helped us improve the manuscript. We have thoroughly considered all comments and suggestions, and made revisions as outlined below (reviewer comments in blue, our response in black bold font).**

**Editor Comments:**

EC1) Your interesting work received two more reviews of which #1 is satisfied ans has no remarks but #2 has still fundamental reservations on several aspects of the methodology of lake volume time series which seem valid points to me. The aspect of the influence of image quality should be addressed and some kind of assessment of the accuracy needs to be given. I also agree that a Pearson coeff R >0.7 is not that a strong correlation. This requires some more argumentation then "Messager et al 2016 used this as well" , i.e. why the authors put the bar on R2~0.5 and why that is acceptable for this global study. These aspects need some more critical discussion in the ms both in methodology and discussion section.

**Thank you so much for your time and effort in handling this manuscript.**

**We have added further analysis regarding image quality. Because the MODIS 8-day composites provide temporally more dense and reliable information, we used the MODIS-derived lake product (Tortini et al., 2020) to investigate the influence of Landsat image quality on the volume time series estimation. Zhao and Gao (2018) regarded images with a contamination ratio ranging from 5% to 95% as 'poor-quality' images and developed an automated fill-gaps method to restore these 'poor-quality' images. We divided reservoir surface water time series (Zhao and Gao, 2018) into four groups (Table S3) according to contamination percentages and validated them against the MODIS-derived water product (Tortini et al., 2020) for 100 lakes. The results (Table S3) show that images with 5-35% contaminated ratio do not affect the temporal accuracy (R=0.87) of lake volume estimation after applying the gap-filling method. The performance drops slightly to R=0.80 when contaminated ratio increases to 35-65% but does not decrease further between 65% and 95%. Overall, the performance of lake volume estimation using gap-filled images (contaminated ratio ranging from 5% to 95%) is the same as using good-quality images, thanks to the gap-filling method.**

**Table S3** The average performance of lake volume estimation using either good quality Landsat images or gap-filling images (original ones with different contamination percentages).

| Contamination Percentages | <5% | 5% - 35% | 35% -65% | 65% - 95% |
|---|---|---|---|---|
| R | 0.86 | 0.87 | 0.80 | 0.80 |
| | 0.86 | | 0.87 | |

**We added this table into the Supplementary and included this analysis in L251-255:**

*"We investigated the influence of Landsat image quality on the volume time series estimation by comparing time series derived from images with different contamination ratios (0~95%) against the MODIS-derived lake product (Tortini et al., 2020). The temporal accuracy slightly decreases as the contamination ratio increases (Table S3). However, the overall performance of lake volume estimation using images with contaminated ratio ranging from 5% to 95% is commensurate to using only good-quality images, thanks to the gap-filling method."*

**We agree that the higher hypsometric correlation we used, the less uncertainties they have. But the trade-off is that using stricter filtering would lead to less data available, especially for a global analysis. The less information used, therefore the less reliable the conclusion. Among the 58 reservoirs with correlations above 0.7, 29 reservoirs have R≥0.9 while 13 have R between 0.7 and 0.8. These data are mainly used for trend analysis, in which we used the annual mean value and total volume in a basin. The uncertainties from the hypsometry will therefore decrease to a much lower level in this temporal and spatial aggregation.**

**There is no strict rule to determine which Pearson correlation threshold between A-L or A-V should be considered before calculating lake volume. Tortini et al. (2020) used R≥0.85 and Busker et al. (2019) used R≥0.9, while Gao et al. (2012) used R≥0.5 (including 0.76 for Port Peck, 0.72 for Sakakawea, 0.83 for Mead and Oahe, and 0.66 for Powell). However, note that the sample size (N) varied between these studies, which is the other variable in addition to the correlation coefficient (R) that affects the interpreted statistical significance via the p value. In our study, the average number of samples (monthly A-L pairs) for each lake is around 166. We calculated significant Pearson correlation threshold using:**

$$t = R \frac{\sqrt{N-2}}{\sqrt{1-R^2}} \quad \text{(S1)}$$

**The result suggests the linear relationship is significant (p<0.01) when R is above 0.19. We also calculated p values using satellite-derived water extents and heights for each lake individually. In all cases, p < 0.01 when R was above 0.18. Therefore, we argue 0.7 is a relatively conservative high correlation in our cases.**

**We added sentences in L131-136 in the Data and Methods Section to explain why we chosen R≥0.7:**

*"The interpretation of Pearson correlation depends on p value and the number (N) of samples. Among these 132 reservoirs, the average number (N) of sample (i.e., the monthly pairs of extent and height) is around 166. We used a significance level of p<0.01 to determine the corresponding Pearson correlation threshold with the t-test (Eq. S1). The result suggested that the linear relationship is significant when R is above 0.19. In this context, we conservatively considered R≥0.7 as evidence of strong correlation. Such a strong correlation between extent and height was found for 58 reservoirs (Group A; Fig. 1)."*

**We add sentences in L426-432 in the Discussion to the selection R≥0.7:**

*"The higher hypsometric correlation we used, the less uncertainties volume estimations would have (Crétaux et al., 2016). We selected correlation threshold of 0.7 in this study, which is lower than Tortini et al. (2020) (R≥0.85) and Busker et al. (2019) (R≥0.9), but higher than Gao et al. (2012) (R≥0.5). The selection of an appropriate correlation threshold can also depend on the purpose of the study. Tortini et al. (2020), Busker et al. (2019) and Gao et al. (2012) aimed to provide accurate measurements for an individual reservoir. Here, our priority is to understand the 32-year volume trend at basin scale. The uncertainties from the individual hypsometry (0.9≥R≥0.7; total 29 reservoirs) therefore average out by temporal (i.e., annual) and spatial (i.e., basin) aggregation."*

**[1] Crétaux, J.-F., Abarca-del-Río, R., Berge-Nguyen, M., Arsen, A., Drolon, V., Clos, G., & Maisongrande, P. (2016). Lake volume monitoring from space. Surveys in Geophysics, 37, 269-305**

EC2) Then I personally found it hard to always follow when it is about numbers of reservoirs, total stored volume or percentages of those. I think it would help the reader if you would report that in a consequent way (e.g. always give number and stored volume) and the percentages of those.

**Thank you for your suggestion. We modified the corresponding sentences and reported number/volume and its percentage in a consistent way in L182-194 in the revised manuscript:**

*"There are 6,862 reservoirs reported in the GRanD database (Lehner et al. 2011), with the total 6,196 km³ reported storage capacity. In this study, we were able to estimate monthly storage dynamics for 6,695 or 97.6% of the total number of reservoirs, with 3,941 km³ or 63.6%of cumulative capacity (Fig. 1). There were only 58 (0.8%) reservoirs for which storage dynamics could be estimated most directly, by a combination of satellite extent and water level observations (Group A), but together they already represent up to 1,394 km³ (22.5%) storage capacity (Fig. 1). The total capacity of the 172 (2.5%) reservoirs not measured constitutes 2,255 km³ (36.4%) of storage capacity. There were 6,637 (96.7%) reservoirs in Group B for which by the geo-statistical approach could be applied, and their total capacity is 2,547 km³ (41.1%). To ensure consistency in the 1984-2015 time series used for long-term trend analysis, we ignored reservoirs with less than 360 months (i.e., 30 years) of Landsat-derived observations or for which more than five years of water extent observations were inter- or extrapolated by Zhao and Gao (2018). Our focus was on interactions between precipitation, streamflow, evaporation and storage in existing reservoirs, rather than the consequences of new impoundments. Therefore, we excluded from consideration all reservoirs that were destroyed, modified, planned, replaced, removed, subsumed or constructed after 1984. This left 4,573 (66.6%) reservoirs available for with combined storage capacity of 2,583 km³ (41.7%) (Fig. 1)."*

**Reviewer #2 Comments:**

I appreciate the authors' efforts on addressing my comments. The revised manuscript reduces parts of my previous concerns. But I still hold my major concerns as some of the responses are not very solid. Please see my comments below.

R2C1) The authors claim that for 5,917 reservoirs that have Landsat observations every month from 1984 – 2015. However, most of the monthly estimates were generated from poor-quality images. I know Zhao and Gao did a comparison between estimated areas and directly observed areas for all reservoirs. They did not provide a comprehensive assessment on the fidelity of the time series for each individual reservoir. This is less a concern for their study as they were interested in aggregate global reservoir areas. I agree that using Zhao and Gao's approach, you can get monthly (or near-monthly) area time series. Here, you attributed storage change in each reservoir case by case. Do the estimates from poor-quality images accurately capture the monthly variability for each reservoir? I think it would be helpful to compare the accuracy of areas from good quality images vs bad. It is seemingly less confident for areas estimated from only a fraction (e.g., 5-10%) of their ROI. Including additional estimates from poor quality images may be fine but also needs to consider the accuracy from these "poor" estimates.

**Thank you for your suggestion. In line with Zhao and Gao (2018), we aggregated reservoir volumes at basin scale before carrying out trending analysis, which reduce uncertainties from individual reservoir. In addition, we did analysis on the influence of Landsat image quality on the volume time series estimation - please see our response to comment EC1.**

R2C2) I am not sure why the author chose to use R instead of R2. A perfect A-V relationship should have a R2 of 1. Busker et al studied 137 lakes and found 58 water bodies having a hypsometric relationship with a R2 > 0.8. I suggested the authors to compare their results with Busker et al. I do not understand why they chose to use R rather than R2 and a lower threshold (0.7). In your studied 132 large reservoirs, 58 water bodies have a hypsometric relationship with a R2 > ~0.5. It seems that your hypsometries have significantly larger uncertainties compared with theirs. As uncertainty in hypsometry critically matters the accuracy of the volume estimate (Cretaux et al. 2016). I am not convinced that their reservoir volume estimates are based on the state-of-art approaches.

Crétaux, J. F., Abarca-del-Río, R., Berge-Nguyen, M., Arsen, A., Drolon, V., Clos, G., & Maisongrande, P. (2016). Lake volume monitoring from space. Surveys in Geophysics, 37(2), 269-305.

**Thank you for this comment. We discussed on why we chosen R≥0.7 in response to the editor, please see our response to comment EC1.**

R2C3) Only a fraction (<3%) of studied reservoirs have observed water levels. I would recommend the authors pay a better attention to the concepts here. A reference volume is a static value, rather than the observed volumes (e.g., by a gauging station). The reported values do not make sense as they cannot justify the fidelity of the volume time series for each reservoir. Why not using observed storages from in-situ stations to validate your estimates? I know at least hundreds of U.S. reservoirs are gauged. As this empirical approach has been applied to the vast majority (>97%) of reservoirs and this approach is

not well validated, I still concern about the quality of the generated datasets. Is your approach remote-sensed based or an empirical based or a mix of them?

**Indeed, the volume is estimated based on a geo-statistical model for the vast majority (in terms of number) of reservoirs (Group B). Specifically, lake depth can be extrapolated from surrounding topography if we have satellite-derived surface water extent observation. The validation results mainly cover lakes from Group B. To validate more reservoirs more comprehensively, we contacted Australian Bureau of Meteorology again and got the updated in-situ stations covering more reservoirs in Australia. We also accessed more in situ reservoir storage records from US Bureau of Reclamation via https://www.usbr.gov/uc/water/hydrodata/. We included these updated validations in L239-245:**

> *"In situ monthly storage records from the US Army Corps of Engineers, US Bureau of Reclamation and Australian Bureau of Meteorology were used for error assessment. There are totally 131 reservoirs with at least 20-year overlapped time series between in situ data and satellite-derived data. We did validation for all these 131 reservoirs (5 for Group A and 126 for Group B). The averaged correlation between observed and estimated volumes is 0.82 (R>0.7 for 82% of the 131 reservoirs). Messager et al. (2016) reported that the symmetric mean absolute percent error (SMAPE) of the geo-statistical model is 48.8% globally. In our study, the average SMAPE between predicted and reference volumes was 32.13%, lower mainly because we adjusted reservoir storage estimates by reported reservoir capacity."*

R2C4) Do the 65 reservoirs cover a fraction of reservoirs with 20-year in-situ data or a subset? I mean you validated on ~1% of reservoirs and recommend justify they are representative. I am surprised that the authors used r to present the accuracy. A perfect r can still be associated with under- or over-estimation. How many of the 65 reservoirs cover the water bodies without water levels?

**In the revised, expanded validation results, we validated overall 131 reservoirs with at least 20-year overlapped time series between in situ data and satellite-derived data (5 for Group A and 126 for Group B). Please see our response to comment R2C3. We provided both R and SMAPE to assess the accuracy of lake volume estimation. However, we indeed more forced on correlation coefficient (temporal accuracy) because this study aims to understand reservoir storage trends, rather than absolute values (L177-179).**

R2C5) I do not understand why you need this assumption: Assuming the estimation method for Group A is more accurate than that for Group B, the latter can also be evaluated against the former. Why not grasp available in-situ data for the validation on Group B?

**Apologies for the confusion. We did use in-situ data to validate Group B. Please see our response to comment R2C3. And this part means we did cross-validation between Group A and Group B. We modified this sentence to clarify this point in L248-249 in the revised manuscript**

> *"In addition, we did cross-validation between Group A and Group B. The results show that 25 of the total 33 overlapping estimated reservoirs show strong agreement (R≥0.9) between the two*

*methods, and the average SMAPE between them is 13.1%. This implies good consistency of reservoir storage estimates from Group A and B. Some cross-validation examples are shown in Fig. 3."*

R2C6) The authors claimed that they explicitly considered the difference between coincidence and causation in our study, which does not seem to be the case. How confident can you attribute the reservoir storage change based on the trending consistency of in-situ river flow and storage? How about in-situ flow only explaining <50% of the storage trend? Additionally, in-situ river flow can be affected by human activities. I am not convinced that the authors examined the causation confidently.

**We agreed that trend consistency is not enough to examine the causation confidently. In this study, we provided four lines of evidence (including analysis of net evaporation and global water withdrawal data) to explore causation (L81-94), rather than only looking at trend consistency. To emphasize this, we summarized the causation analysis in L486-494 in the Conclusion.**

> *"We provided four lines of evidence to explore which factor (precipitation, net evaporation, or dam (demand-related) water releases) drives the global reservoir storage trends. First, we found trend consistency between precipitation, streamflow and reservoir storage. Second, we found robust temporal correlation between precipitation, streamflow and reservoir storage. Third, we inferred the role of human activity based on the reservoir water balance equation: because we found changes in net evaporation only accounted for a small fraction of reservoir volume changes, together with the first two lines of evidence, we can infer that dam (demand-related) water releases are less likely to be the main driver of storage changes. Fourth, we examined water use data and did not find that increasing water use corresponded to deceasing reservoir storage, or vice versa, in the majority of basins. Therefore, we conclude that reservoir volume changes are dominated by (multi-decadal) precipitation changes."*

R2C7) As stated earlier, large reservoirs are highly regulated by humans. The proposed approaches have limited capacity to capture the human reservoir regulations, which could falsely sign the dominant importance of natural climate variability.

**We agreed that this study has limited capacity to understand the influence of human reservoir regulations as there is no available global dam release data. However, we used an indirect approach to infer the role of human activity (L356-370). Please see our response to comment R2C6. We also argued that human interventions can affect seasonal variability in reservoirs (Cooley et al., 2021) but not necessarily multi-decadal trends (L444-448).**

R2C8) Decreased evaporation could be a result of decreased water surface. Comparing the trending directions between these two does not tell whether E is the driver or not.

**In terms of net evaporation, we did not compare trend consistency. We can calculate how much of storage changes can be explained by changes in net evaporation. The results showed that changes in net evaporation accounted for well below 10% of the overall trends in storage (Fig. 10). This means there is little influence of net evaporation on reservoir storage changes.**

R2C9) Why not using a water mass balance approach? Isn't more robust compared with the seemingly simplified comparisons of trending directions?

**We cannot use the water mass balance approach directly as there is no in situ or simulated inflow and water release data for all individual reservoirs. Instead, we analyzed the influence of climate variability and human activity on the basin-scale reservoir storage changes based on reservoir water balance (Eq. 9) in an indirect way (L356-370). Please see our response to comment R2C6.**

R2C10) "There were only 58 reservoirs for which storage dynamics could be estimated most directly, by a combination of satellite extent and water level observations (Group A), but together they already account for 25.5% of combined global reservoir capacity (Fig. 1). The total capacity of the 193 reservoirs not measured constitutes 36.4% of global capacity. There were 6,611 reservoirs in Group B for which by the geo-statistical approach could be applied, and these contribute 41.1% to total global capacity."

Aren't your conclusions based on the large number of reservoirs which storage estimates have the largest uncertainty due to the empirical method?

**We have performed validation for both Group A and Group B and added more validation results for Group B in the revised manuscript. The result indicated that the generated dataset has a robust temporal accuracy, which is sufficiently reliable for trend analysis. Please see our response to comment R2C3. We also performed cross-validation between Group A and Group B. The result showed that they have a comparable level of temporal accuracy. Please see our response to comment R2C5.**

R2C11) "Our focus was on interactions between precipitation, streamflow, evaporation and storage in existing reservoirs, rather than the consequences of new impoundments. Therefore, we excluded from consideration all reservoirs that were destroyed, modified, planned, replaced, removed, subsumed or constructed after 1984."

As you focus on old reservoirs, does sedimentation affect the estimated storage? It seems to be a significant issue for old reservoirs.

**We agree that sedimentation can also contribute to the decrease of reservoir water storage but the effect of sedimentation on our global 32-year analysis will be small, according to Wisser et al. (2013) (L437-440).**

---

## Author Response (AR3)

**Response to Reviewers**

**''Remotely sensed reservoir water storage dynamics (1984-2015) and the influence of climate variability and management at global scale'' by Jiawei Hou et al.**

**We thank the editor and the reviewer for the thoughtful comments and helpful suggestions. We have thoroughly considered all comments and suggestions, and made revisions as outlined below (reviewer comments in blue, our response in black bold font).**

Editor Comments:

EC1) you received another review of a reviewer who looked at the manuscript before. The reviewer still has fundamental issues especially on the validity of the estimated storages. I am not sure if all these uncertainties can be solved and I think the inherent uncertainties in reservoir area - reservoir storage should not block the publication. I do request you explicitly mention this uncertainty in both abstract and conclusion. It is a fundamental aspect and also for readers who only study abstract or conclusions, I think, this should be clear.

**Thank you so much again for your time and effort in handling this manuscript. To clarify the uncertainties in reservoir area and storage estimates, we added sentences in L20-21 in Abstract:**

> *"Uncertainty in the analysis can come from, among others, the relatively low Landsat imaging frequency for parts of the Earth and the simple geo-statistical bathymetry model used."*

**In L495-497 in Conclusion:**

> *"For reservoirs with water extent data only, storage was estimated from the surrounding topography using a geo-statistical model. This approach introduces uncertainty but is inevitable as lake bathymetry data based on surveys are typically unavailable, at least in the public domain."*

**In L502-503 in Conclusion:**

> *"With lower-frequency observations, Landsat may not always have fully or accurately captured the storage variability for each reservoir, which can have had an effect on trend analysis."*

EC2) The effect of the size of the reservoir on the storage trends (a few large reservoirs vs many small ones) needs to be addressed as well. I look forward to the revised version explicitly discussing the reviewers remarks and revise the ms accordingly.

**We agree with the editor and reviewer #1 that total storage trend in a basin can be dominated by a few large reservoirs. However, we should not ignore the cumulative effects of a large number of smaller reservoirs. We compared the trend directions of total storage in all reservoirs, top three largest reservoirs, and the rest of small reservoirs in 42 basins that have more than 20 reservoirs (overall 4,003 reservoirs). The results indicated our trends analysis was not dominated by a few large reservoirs for all basins. We added sentences in L354-360 to present this analysis:**

*"To understand the influence of the reservoir size distribution on the total basin storage trends, we compared the trend directions of total storage in all reservoirs, the top-three largest reservoirs, and the remaining small reservoirs, respectively. We did this for 42 basins with more than 20 reservoirs (4,003 reservoirs in total). Combined storage in these three groups all showed the same trend direction in 27 (62.8%) of basins. The trend in the combined storage for all reservoirs had the same direction as that for the largest few reservoirs for 8 more basins, and the same direction as the combined remaining smaller reservoirs for another 8 basins. This indicates that the largest reservoirs do not always dominate combined total storage dynamics."*

**Reviewer #1 Comments:**

Hou et al tried to add additional analysis and clarification in response to my concerns on both the quality of their dataset on global reservoir storage trends and the confidence of their conclusion on the driving factors. Overall, their responses and revision helped improve the quality and clarity of their manuscript. However, the provided evidence/explanation is not sufficient enough because some analysis and arguments seem to be incomplete and sometimes misleading.

This is a very interesting study and in line with recent efforts on understanding long-term changes in lake water storage worldwide. The study potentially contributes to providing a global storage database for thousands of reservoirs and identifying the drivers of the changes. But I also found there is a lot of uncertainties affecting their analysis and the potential usage of their dataset. This is why I suggested them provide a comprehensive evaluation of their estimated storage and trends. The authors did additional experiments on justifying the impact of poor quality images and the storage data for reservoirs without water level data.

**We thank the reviewer for the time and effort in reviewing this manuscript again and providing these thoughtful comments.**

R1C1) In terms of the impact of image quality, the authors did an additional comparison with the MODIS-derived time-varying lake dataset. The resolution of the MODIS is 250 m. How can a 250-m product be used to validate your estimates at a much finer resolution (30-m)? As seen in many relevant papers, Landsat products were used to validate the MODIS-derived products. It never makes sense to do it in a reserve way. In particular, MODIS is very limited capacity to monitor the area changes in relatively small reservoirs (Gao et al. 2012).

"Gao, H., Birkett, C., & Lettenmaier, D. P. (2012). Global monitoring of large reservoir storage from satellite remote sensing. Water Resources Research, 48(9)."

**We agree that it does not make sense to use low spatial resolution data to validate high resolution one. We would like to emphasise here that we intended to compare, rather than validate, our estimates against Tortini et al. (2020)'s product, based on which we tried to understand the impact of Landsat image quality on storage estimates. Especially, the MODIS 8-day composites used in Tortini et al. (2020) provide temporally denser observations, which is an important feature that can be used to investigate low temporal resolution Landsat 'poor-quality' images.**

**The higher imaging frequency of MODIS does not improve its quality per se, but does provide more opportunity for temporal compositing.**

R1C2) The authors also conducted additional validation on reservoir storage estimates from an empirical model. As I pointed out earlier this empirical model is designed for estimating the total volume rather than the volume change. The shape of lake bathymetry is dependent on a lot of factors and cannot simply be extrapolated from shoreline slope. Using a simple empirical model seems to be insufficient for estimating storage variability and trends. The authors argued that their estimates are reasonable based on the correlations with the in-situ storage time series. But this is not a direct validation. As you used the simple area-volume scaling, how the estimated storage trends agree with the trends from in-situ storage time series? The authors did not respond to my comment that the estimated storage time series for reservoirs with level data seems to be worse than recent estimates given lower R2 values. Thus, I really would like to see the solid evidence supporting that their empirical models can be used to estimate storage trends in reservoirs without level observtions as I think this is one of their major contributions.

**Thank you for your suggestion. We validated significant positive trends, no trends, and significant negative trends derived from our reservoir storage estimates against these from in situ storages and MODIS-observed volumes (Tortini et al., 2020). The results showed that there was no opposite trend in any of the cases, and the trend significance level (i.e. $p<0.05$ or not) agreed for 93% and 91% of reservoirs, respectively (Table 1).**

**Table 1 Comparison of significant trends in reservoir storage reconstruction against in situ storages and MODIS-based estimates.**

|  |  | Our estimates | | |
|---|---|---|---|---|
|  |  | Decreasing | No Trend | Increasing |
| In situ storage | Decreasing | 12 | 0 | 0 |
|  | No Trend | 4 | 45 | 0 |
|  | Increasing | 0 | 1 | 5 |
| MODIS-based storage | Decreasing | 8 | 1 | 0 |
|  | No Trend | 4 | 59 | 1 |
|  | Increasing | 0 | 3 | 24 |

R1C3) To address the limited confidence of the attribution analysis, they did additional correlation analysis as well as considered the human water use data. It is still confusing to me that they did the attribution analysis at a large basin scale (Level 3 of HydroBasins) rather than at each reservoir basin. What does it really mean for the consistency and correlations at this large basin scale? Storage trends at this basin level can be largely attributed to a few largest reservoirs and the vast majority of smaller reservoirs barely contribute. Does the identified driver really represent the dominant influence for all reservoirs in that basin? Each reservoir may be operated differently, and thus using large-basin-level water use data can underestimate the human impact. The statements like "Many of the observed

reservoir changes could be explained by changes in precipitation and river inflows" (line 18-19) are not fully supported by the results.

**It would be a major challenge to define the catchment boundary for each reservoir using remote sensing or modelling globally. We argue that combined total storage provides a reasonable representation for the whole basin, and conducting analysis at a large basin scale can reduce uncertainties from data used in this study (e.g., precipitation, streamflow, reservoir area/storage). In addition, we demonstrated that our trends analysis was not always dominated by one or a few large reservoirs. See our response to EC2.**

R1C4) They stated the impact of sedimentation on the 32-year analysis is small given the only 5% of global reservoir water storage being lost to sediment over a century (Wisser et al. 2013). However, based on recent studies, the sedimentation rate does not seem to be that small. For example, Jaia et al summarized the Anthropocene sediment budget for Earth and found the sedimentation rate in global reservoirs is pretty high over recent decades (e.g., 60 Gt per year in 2010). Additionally, the sedimentation rates in the vast majority of smaller reservoirs can be even larger. How sedimentation impacts their analysis and conclusions remains unclear.

"Syvitski, J., Ángel, J. R., Saito, Y., Overeem, I., Vörösmarty, C. J., Wang, H., & Olago, D. (2022). Earth's sediment cycle during the Anthropocene. Nature Reviews Earth & Environment, 3(3), 179-196."

**We accept there can be uncertainties in global reservoir sedimentation estimation, as different studies came to different sedimentation rates, for example 5% in 100 years (Wisser et al. 2013) vs. 1% in 2010 (Syvitski et al., 2022). We acknowledge that sedimentation could be a factor affecting reservoir trends analysis in L451-454:**

> *"There are studies showing higher sedimentation rates (e.g., Syvitski et a., 2022), so the impact of sedimentation on reservoir trend analysis cannot be discounted entirely. Thus, decreasing storage volume could be exacerbated by sedimentation, while increasing storage volumes could potentially be (partly) explained by it."*

Specifically comments:

R1C5) 150 – 155: The statement here is not precise. Please note that bathymetry is different from the mean depth. While Messager et al. developed a geostatistical approach to estimate the mean depth, the application of this approach for bathymetry estimates needs to be examined.

**Thank you, we changed "bathymetry" to "mean depth" in the revised manuscript.**

R1C6) 244 SMAPE is still pretty large (32.13%). How this uncertainty affects the estimated trend?

**We compared satellite-derived storage trends against in situ storage trends to investigate storage trend uncertainty; see our response to R1C2.**

R1C7) Figure 4. I suggest a map of storage trend uncertainty to be added.

**See response to R1C6.**

R1C8) 260: Could you provide an uncertainty for the aggregate global storage trend?

**See response to R1C6.**

R1C9) 365 – 369: The reference Wang et al. does not seem to support the statement here as they, i.e., Hou et al., focused on the impact of human activities on reservoirs (e.g., Three Gorges reservoir) rather than its downstream lakes.

**We cited two examples (including Wang et al., 2017) here to show that some recent studies found that climate variability dominates the changes in lakes and rivers, rather than water and land management (L377-381). These provide supporting evidence for our interpretation, in terms of global surface water (i.e., river, lake and reservoir) changes.**

R1C10) 388-389: This conclusion that climate trends rather than water withdrawals are primarily responsible for the observed trends in reservoir storage is not fully supported by the presenting evidence.

**We tried our best to use different kinds of available global data to infer this conclusion. But we accept that it would be preferable to find more direct evidence (e.g., in situ dam release records worldwide) to verify our conclusion (L474-477). Unfortunately, these are not currently available for the vast majority of reservoirs.**